# Abnormal expression and the significant prognostic value of aquaporins in clear cell renal cell carcinoma

**Mingrui Li, Minxin He, Fangshi Xu, Yibing Guan, Juanhua Tian, Ziyan Wan, Haibin Zhou, Mei Gao, Tie Chong** *

Department of Urology, The Second Affiliated Hospital, Xi'an Jiaotong University, Xi'an, Shaanxi Province, China

* chongtie@mail.xjtu.edu.cn

**Data Availability Statement:** We have uploaded our data set to a repository. The repository is BioStudies. The accession number is S-BSST812.

## Abstract

Aquaporins (AQPs) are a kind of transmembrane proteins that exist in various organs of the human body. AQPs play an important role in regulating water transport, lipid metabolism and glycolysis of cells. Clear cell renal cell carcinoma (ccRCC) is a common malignant tumor of the kidney, and the prognosis is worse than other types of renal cell cancer (RCC). The impact of AQPs on the prognosis of ccRCC and the potential relationship between AQPs and the occurrence and development of ccRCC are demanded to be investigated. In this study, we first explored the expression pattern of AQPs by using Oncomine, UALCAN, and HPA databases. Secondly, we constructed protein-protein interaction (PPI) network and performed function enrichment analysis through STRING, GeneMANIA, and Metascape. Then a comprehensive analysis of the genetic mutant frequency of AQPs in ccRCC was carried out using the cBioPortal database. In addition, we also analyzed the main enriched biological functions of AQPs and the correlation with seven main immune cells. Finally, we confirmed the prognostic value of AQPs through GEPIA and Cox regression analysis. We found that the mRNA expression levels of AQP0/8/9/10 were up-regulated in patients with ccRCC, while those of AQP1/2/3/4/5/6/7/11 showed the opposite. Among them, the expression differences of AQP1/2/3/4/5/6/7/8/9/11 were statistically significant. The differences in protein expression levels of AQP1/2/3/4/5/6 in ccRCC and normal renal tissues were consistent with the change trends of mRNA. The biological functions of AQPs were mainly concentrated in water transport, homeostasis maintenance, glycerol transport, and intracellular movement of sugar transporters. The high mRNA expression levels of AQP0/8/9 were significantly correlated with worse overall survival (OS), while those of AQP1/4/7 were correlated with better OS. AQP0/1/4/9 were prognostic-related factors, and AQP1/9 were independent prognostic factors. In general, this research has investigated the values of AQPs in ccRCC, which could become new survival markers for ccRCC targeted therapy.

**Funding:** The author(s) received no specific funding for this work.

**Competing interests:** The authors have declared that no competing interests exist.

## Introduction

Renal cell carcinoma (RCC), the most common malignant tumor of the kidney, is diagnosed with more than 400,000 people and is associated with over 175,000 deaths worldwide per year [1]. The most common form of RCC is the clear cell renal cell carcinoma (ccRCC), which accounts for around 70% of cases [2]. Although most incidentally detected lesions are small low-grade tumors, up to 17% of all RCC patients have distant metastases at the time of diagnosis [3]. The early-stage RCC is better treated by resection, but over 30% of patients develop metastatic progression after treatment [4]. Because of the resistance of kidney cancer to chemotherapy or radiotherapy and the limited response to immunotherapy, metastatic RCC presents a particular therapeutic challenge to clinicians. Although several multi-targeted tyrosine kinase inhibitors have been assessed for patients, most patients experience partial responses or stabilization of disease and only a few patients show complete response to treatment [5]. So there is an urgent need for new prognostic markers and therapeutic targets.

Aquaporins (AQPs), a family of small hydrophobic proteins, are incorporated into cell membranes and are expressed in all living organisms. These integral proteins have been found to form transmembrane water channels that play critical roles in controlling the water flow into and out of cells and permeating other small solutes such as anions, urea, and glycerol [6]. To date, 13 AQPs members have been identified in humans, including AQP0–AQP12. Among them, AQP0 is previously known as major intrinsic protein (MIP) [7]. AQP12 is divided into two subtypes called AQP12A and AQP12B [8]. By adjusting cell permeability, mediating glycerol translocation, taking part in some specific signaling pathways and other biological functions, AQPs are not only expressed in various epithelial and nonepithelial tissues for regulating rapid water movement but also actively implicate and regulate a variety of cancers [9]. For instance, previous studies had identified the expression and prognostic value of AQPs in pancreatic cancer and gastric cancer [10, 11]. Moreover, prior research revealed that the level of AQP1 in urine is a sensitive diagnostic marker for ccRCC and papillary renal cell carcinoma [12].

However, the prognostic value of AQPs in ccRCC has not been thoroughly dissected, and the roles of different AQPs members in the initiation and development of ccRCC are still unclear. In this study, we integrated data of cancer gene expression from different online databases and performed a variety of bioinformatics analyses. We aimed to analyze the differences in AQPs mRNA expression between ccRCC and normal kidney tissues. Moreover, we investigated the influences of AQPs expression on ccRCC prognosis and the predicted functions of AQPs and their similar genes. This study will help us find new targets for the treatment of ccRCC.

## Materials and methods

### Ethics statement

All the datasets were retrieved from the published literature, so all written informed consent had been previously obtained.

### Oncomine analysis

Oncomine is currently the world's largest tumor gene chip database aiming to mine cancer genetic information [13]. The Oncomine online database was used to analyze the mRNA expression levels of AQPs in different human tumors. We compared the expression of AQPs between the ccRCC and normal kidney tissues, 'AQP0/1/2/3/4/5/6/7/8/9/10/11/12A/12B' were selected as the keywords in search, and 'Clear Cell Renal Cell Carcinoma vs. Normal Analysis'

was chosen as 'Analysis Type'. The threshold was set to gene rank, 10%; fold change, 2; and P-value, 0.01.

## UALCAN analysis

As a comprehensive online analysis tool, the UALCAN database is based on OMICS data (TCGA, MET500, and clinical proteomic tumor analysis consortium). Meanwhile, convenient functions such as pan-cancer gene expression analysis and specific gene expression level analysis are provided through this platform [14]. In this study, AQPs were used as the basis to analyze the expression levels between ccRCC and normal kidney tissue, 'AQP0/1/2/3/4/5/6/7/8/9/10/11/12A/12B' were selected as the keywords in search, and 'Kidney renal clear cell carcinoma' was chosen as 'TCGA dataset'. Student's t-test was used to generate a p-value. The p-value cutoff was 0.05.

## Cancer Genome Atlas (TCGA) database

The Cancer Genome Atlas (TCGA) molecularly characterizes over 20,000 primary cancer and matches normal samples spanning 33 cancer types. TCGA generates over 2.5 petabytes of genomic, epigenomic, transcriptomic, and proteomic data, which has already led to improvements in the ability on diagnose, treatment, and prevention for cancer [15]. In this study, the clinical data of 530 ccRCC patients on the Genomic Data Commons Data Portal website were downloaded, and the data of 4 patients were excluded because of the absence of follow-up data. Finally, univariate and multivariate Cox regression analysis of the overall survival (OS) and disease free survival (DFS) of ccRCC patients were performed on the remaining 526 sample data. The p-value <0.05 was regarded as statistically significant.

## GEPIA analysis

Gene Expression Profiling Interactive Analysis (GEPIA), a developed interactive web server, uses a standard processing pipeline to analyze the RNA sequencing expression data of 9,736 tumors and 8,587 normal samples from the TCGA and the GTEx projects [16]. In this research, the " Single Gene Analysis" module was used to analyze the relationship between mRNA expression levels of AQPs and individual cancer stages of ccRCC patients, and 'KIRC' was chosen as 'Datasets'. The p-value < 0.05 was considered statistically significant. Student's t-test was used to generate a p-value for pathological stage analysis.

## cBioPortal analysis

cBioPortal is a relatively comprehensive online analysis tool that can conduct data exploration and analysis of multidimensional cancer genomics and clinical data [17]. In this study, AQPs-related mutations, OS, DFS, and co-expression were analyzed based on 538 ccRCC samples in the cBioPortal database (TCGA, Firehose Legacy). The mRNA expression z-scores (RNA Seq V2 RSEM) were obtained using a z-score threshold of ± 2.0.

## GeneMANIA analysis

GeneMANIA is a prediction server for finding other genes that are related to a set of input genes, using an enormous set of functional association data [18]. Association data includes protein and genetic interactions, pathways, co-expression, co-localization, and protein domain similarity. In the experiment, the GeneMANIA was used to analyze the functional relationship between AQPs and related genes. 'AQP1/2/3/4/5/6/7/8/9/10/11/12A/12B and MIP' were selected as the keywords in the search.

## STRING11.0 analysis

STRING, as a platform for searching and interpreting conserved patterns in genome organization, aims to find the functional relationship between given genes, and use potential functions to analyze the results comprehensively and produce protein-protein interaction (PPI) network visually [19]. STRING was used to create a PPI network diagram of AQPs and analyze the relationship between them. The 'Multiple proteins' module was chosen, 'AQP1/2/3/4/5/6/7/8/9/10/11/12A/12B and MIP' were selected as the keywords in search and 'Homo sapiens' was chosen as 'Organism'.

## Metascape analysis

Metascape is an online analysis tool based on the OMICS database. It has functions such as functional enrichment, interactome analysis, gene annotation, and membership search, and it can analyze and annotate given genes [20]. The Metascape's express analysis function was used to further verify and analyze the enrichment relationship between AQPs and adjacent genes. 'AQP0/1/2/3/4/5/6/7/8/9/10/11/12A/12B' were selected as the keywords in search, and 'H. sapiens' was chosen as 'Input as species' and 'Analysis as species'.

## HPA analysis

The Human Protein Atlas (HPA) database is an online resource database containing an extensive amount of protein expression data and high-resolution images of human normal cells and cancer cells [21]. Translational-level validation of AQPs was carried out using the HPA database. The following antibodies were used: MIP (Anti-MIP polyclonal antibody, Atlas Antibodies, Cat# HPA014940), AQP1 (Anti-AQP1 antibody, Atlas Antibodies, Cat# HPA019206), AQP2 (Anti-AQP2 polyclonal antibody, Atlas Antibodies, Cat# HPA046834), AQP3 (Anti-AQP3 polyclonal antibody, Atlas Antibodies, Cat# HPA014924), AQP4 (Anti-AQP4 polyclonal antibody, Atlas Antibodies, Cat# HPA014784), AQP5 (Anti-AQP5 polyclonal antibody, Atlas Antibodies, Cat# HPA065008), AQP6 (Anti-AQP6 polyclonal antibody, Atlas Antibodies, Cat# HPA015278), AQP7 (pending analysis), AQP8 (Anti-AQP8 polyclonal antibody, Atlas Antibodies, Cat# HPA046259), AQP9 (Anti-AQP9 Antibody, Atlas Antibodies, Cat# HPA074762), AQP10 (Anti-AQP10 polyclonal antibody, Atlas Antibodies, Cat# HPA065947), AQP11 (pending analysis), AQP12A (Anti-AQP12A polyclonal antibody, Atlas Antibodies, Cat# HPA042216), AQP12B (Anti-AQP12A polyclonal antibody, Atlas Antibodies, Cat# HPA042216). The classification criteria for protein expression levels was based on the fraction of stained cells (0%, <25%, 25%-75%, >75%).

## TIMER analysis

Tumor Immune Estimation Resource (TIMER) is a comprehensive resource for systematic analysis of immune infiltration of various cancer types [22]. The correlations between the expression of AQPs and the infiltration level of each type of immune cells (B cells, CD4 + T cells, CD8 + T cells, neutrophils, macrophages, and dendritic cells) were analyzed through the spearman correlation test. 'AQP1/2/3/4/5/6/7/8/9/10/11/12A/12B and MIP' were selected as the keywords in search, 'KIRC' was chosen as 'Cance Types' and 'B Cell, CD8+ T Cell, CD4+ T Cell, Macrophage, Neutrophil and Dendritic Cell' were selected as 'Immune Infiltrates'.

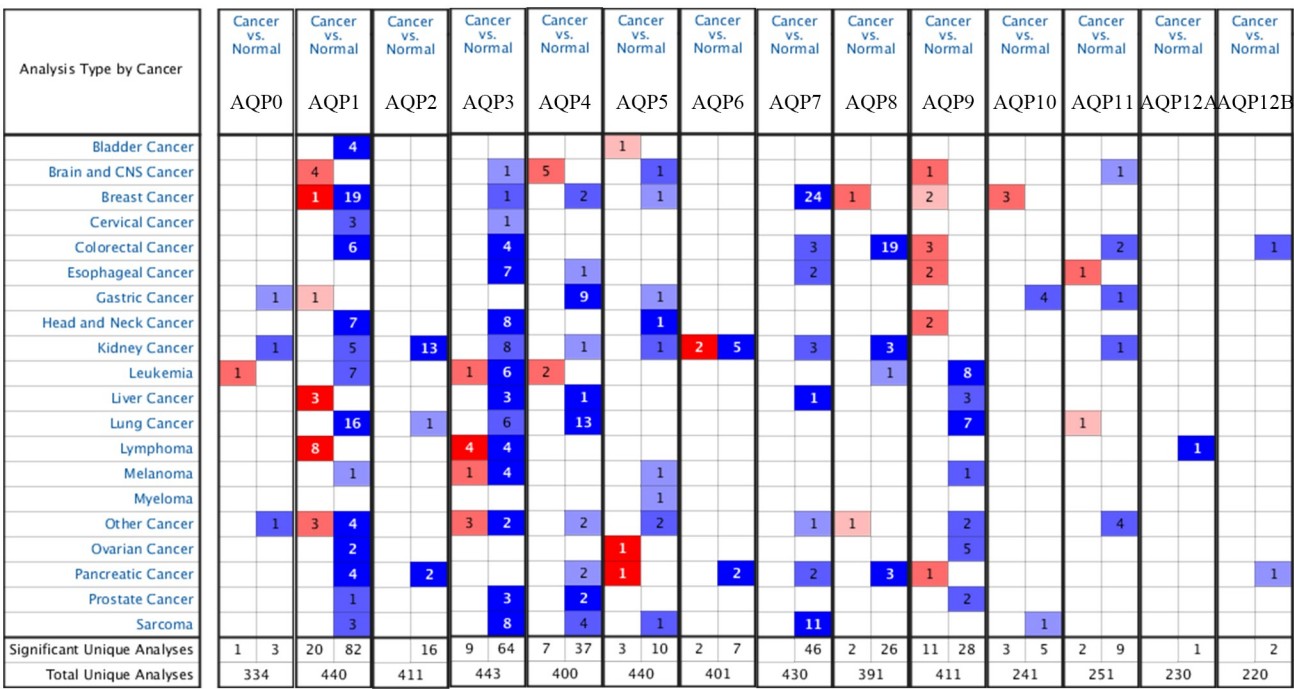

**Fig 1. Transcriptional expression of AQPs in 20 different types of cancer diseases (Oncomine).** The numbers in the small boxes represent the number of datasets with statistically significant mRNA differences of the target genes, in which red represents over-expression and blue represents down-regulated expression. Differences in transcriptional expression between ccRCC and normal kidney tissues were compared by students' t-test. The cut off of p-value and fold change were as following: p-value: 0.01, fold change: 2, gene rank: 10%, data type: mRNA.

## Results

### Expression levels of AQPs in patients with ccRCC

Firstly, we searched and selected AQPs in 20 different tumors through the Oncomine database and compared the mRNA expression levels between ccRCC and the corresponding normal tissues (Fig 1). The results showed that the expression levels of AQP0/1/2/3/4/5/7/8/11 in kidney cancer were significantly lower than those in normal tissues, while the expression level of AQP6 was up-regulated or down-regulated in different test results, which was inconsistent. For AQP9/10/12A/12B, no test data could provide suitable results under the established screening conditions. Next, we analyzed the specific data (Table 1) [23–28]. Among them, Beroukhim conducted experiments on non-hereditary and hereditary ccRCC and found that the expression levels of AQP1/2/3 in ccRCC were significantly lower than those in normal kidney tissues. The fold changes were -4.503, -11.212, -12.336, -3.699, and -2.391, respectively. The Gumz dataset showed that the expression levels of AQP2/6/7 in ccRCC were significantly reduced, with fold changes being -5.59, -4.140, and -9.610, respectively. In ccRCC, Lenburg found that the expression levels of AQP2/7 were down-regulated compared with normal tissues, and the fold changes were -2.567 and -2.12, respectively. The results of Yusenko's dataset showed that the expression levels of AQP2/6 were related to those of normal tissues were reduced, with fold changes of -13.183 and -8.330. The Jones dataset proved that the expression levels of AQP2/8 were down-regulated in ccRCC, and the fold changes were -4.592 and -3.097. Compared with normal tissues, the data of Higgins proved that the expression level of AQP3 in ccRCC was significantly reduced, and the fold change was -3.8889. It could be seen that the RCC-related datasets proved that AQP6 had a low expression in ccRCC. In summary, seven

**Table 1.  Significant changes of AQPs expression in transcription level between ccRCC and normal kidney tissues (Oncomine).**

| Types of ccRCC VS. Kidney | Fold Change | p-value | t-test | Ref |
|---|---|---|---|---|
| AQP1 | | | | |
| Non-Hereditary Clear Cell Renal Cell Carcinoma | -4.503 | 1.70E-05 | -4.751 | Beroukhim Renal [23] |
| AQP2 | | | | |
| Clear Cell Renal Cell Carcinoma | -5.590 | 2.93E-11 | -13.694 | Gumz Renal [24] |
| Clear Cell Renal Cell Carcinoma | -2.567 | 5.55E-6 | -6.779 | Lenburg Renal [25] |
| Clear Cell Renal Cell Carcinoma | -13.183 | 0.002 | -5.550 | Yusenko Renal [26] |
| Clear Cell Renal Cell Carcinoma | -4.592 | 1.46E-10 | -10.233 | Jones Renal [27] |
| Non-Hereditary Clear Cell Renal Cell Carcinoma | -11.212 | 1.87E-08 | -12.468 | Beroukhim Renal [23] |
| Hereditary Clear Cell Renal Cell Carcinoma | -12.336 | 4.06E-08 | -13.460 | Beroukhim Renal [23] |
| AQP3 | | | | |
| Clear Cell Renal Cell Carcinoma | -3.889 | 4.56E-6 | -7.538 | Higgins Renal [28] |
| Non-Hereditary Clear Cell Renal Cell Carcinoma | -3.699 | 8.93E-10 | -8.449 | Beroukhim Renal [23] |
| Hereditary Clear Cell Renal Cell Carcinoma | -2.391 | 9.75E-07 | -5.931 | Beroukhim Renal [23] |
| AQP6 | | | | |
| Clear Cell Renal Cell Carcinoma | -4.140 | 1.18E-11 | -14.586 | Gumz Renal [24] |
| Clear Cell Renal Cell Carcinoma | -8.330 | 1.01E-4 | -7.220 | Yusenko Renal [26] |
| AQP7 | | | | |
| Clear Cell Renal Cell Carcinoma | -9.610 | 4.45E-07 | -9.447 | Gumz Renal [24] |
| Clear Cell Renal Cell Carcinoma | -2.102 | 3.26E-04 | -4.863 | Lenburg Renal [25] |
| AQP8 | | | | |
| Clear Cell Renal Cell Carcinoma | -3.097 | 2.52E-30 | -29.476 | Jones Renal [27] |

different datasets proved that the expression levels of AQP1/2/3/6/7/8 in ccRCC were significantly lower than those of normal kidney tissues.

We also used the UALCAN database to evaluate the differences in the expression levels of AQPs between ccRCC and normal tissues. The dataset used 533 tumor samples and 72 normal tissue samples. The results showed that, except for AQP0/10/12A which had no statistical data, the expression levels of the other eleven AQPs members were statistically significant compared with normal tissues, of which AQP1 (<1E-12), AQP2 (1.11E-16), AQP3 (1.63E-12), AQP4 (5.56E-07), AQP5 (5.75E-03), AQP6 (7.01E-04), AQP7 (2.34E-12), AQP11 (6.57E-07). The expression of AQP8 (4.00E-04), AQP9 (1.58E-05), AQP12B (7.02E-03) were up-regulated (Fig 2).

After evaluating the mRNA transcription and expression of AQPs in ccRCC, to further understand the differences in protein levels of AQPs, we evaluated the expression of AQPs through HPA. We systematically screened the available immunohistochemical images of all available proteins shown in the database. Compared with the ccRCC tissues, several proteins in normal kidneys had been shown to have different expression levels (Fig 3). AQP1/4 were moderately expressed in normal tissues and lowly expressed in ccRCC, AQP0/2/3/6/8/9 had low, medium, or high protein expression levels in normal kidney tissues, but no obvious expression in ccRCC. AQP5/10/12A&B were expressed in normal tissues and the protein expression levels in tumor tissues were not detected (AQP7/11 were not displayed because there were no relevant information in the database). In general, most AQPs members (AQP1/2/3/4/5/6/7/8/9/11/12B) had significant differences in the levels of mRNA expression or protein transcription in ccRCC compared to normal kidney tissues. Then the correlations between the mRNA expression levels and the clinicopathological stages of ccRCC were analyzed, respectively (Fig 4). We found that AQP1 (5.83e−07), AQP2 (0.0146), AQP4 (0.000382),

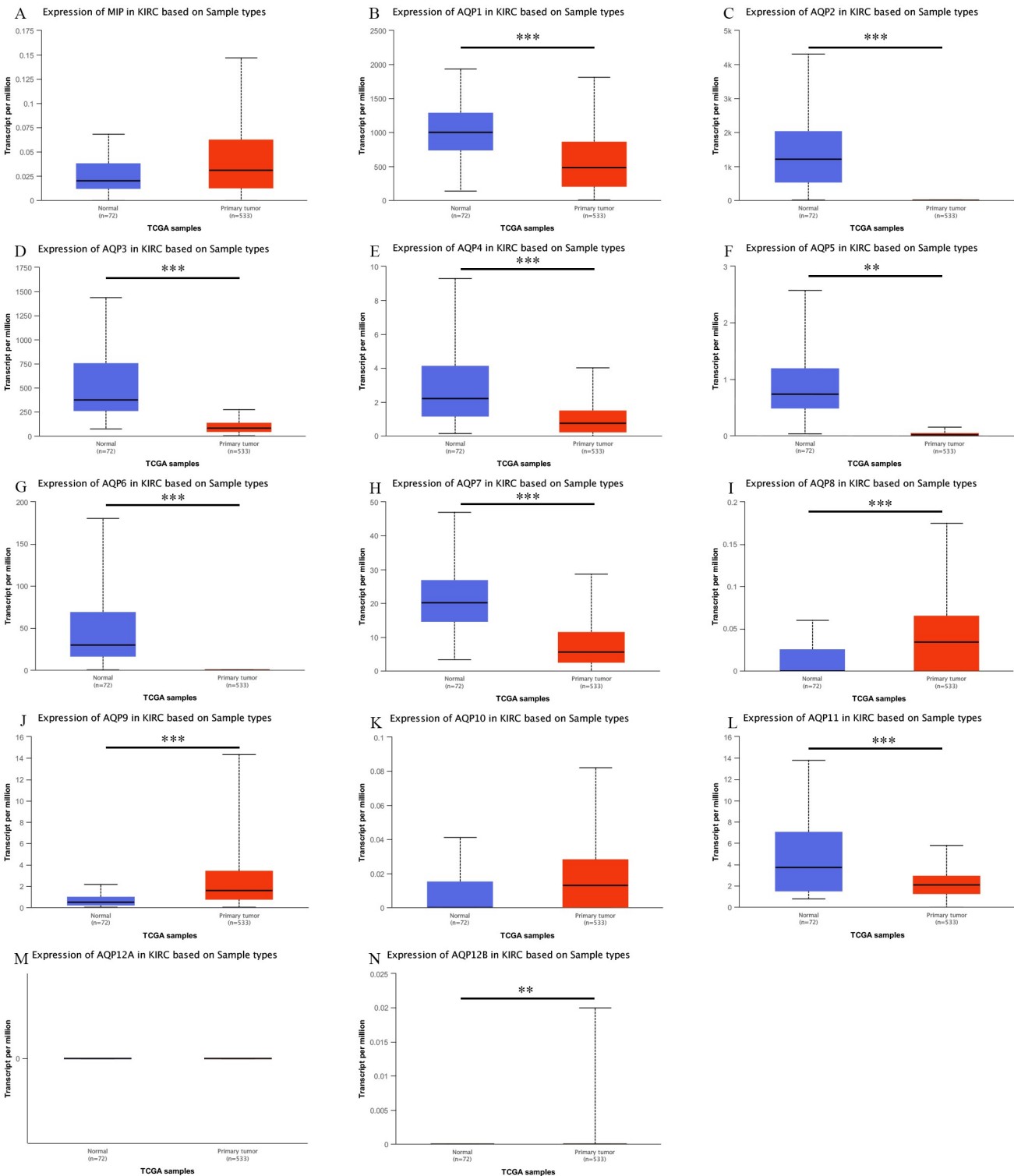

**Fig 2. mRNA expression of AQPs in ccRCC and adjacent normal kidney tissues (Ualcan).** mRNA expressions of AQP1/2/3/4/5/6/7/11 in ccRCC were significantly reduced while the expressions of AQP8/9/12B were significantly elevated. The p-value was set at 0.05. (A) Expression of MIP in KIRC based on Sample types. Expression of (B) AQP1, (C) AQP2, (D) AQP3, (E) AQP4, (F) AQP5, (G) AQP6, (H) AQP7, (I) AQP8, (J) AQP9, (K) AQP10, (L) AQP11, (M) AQP12A, (N) AQP12B in KIRC based on Sample types.

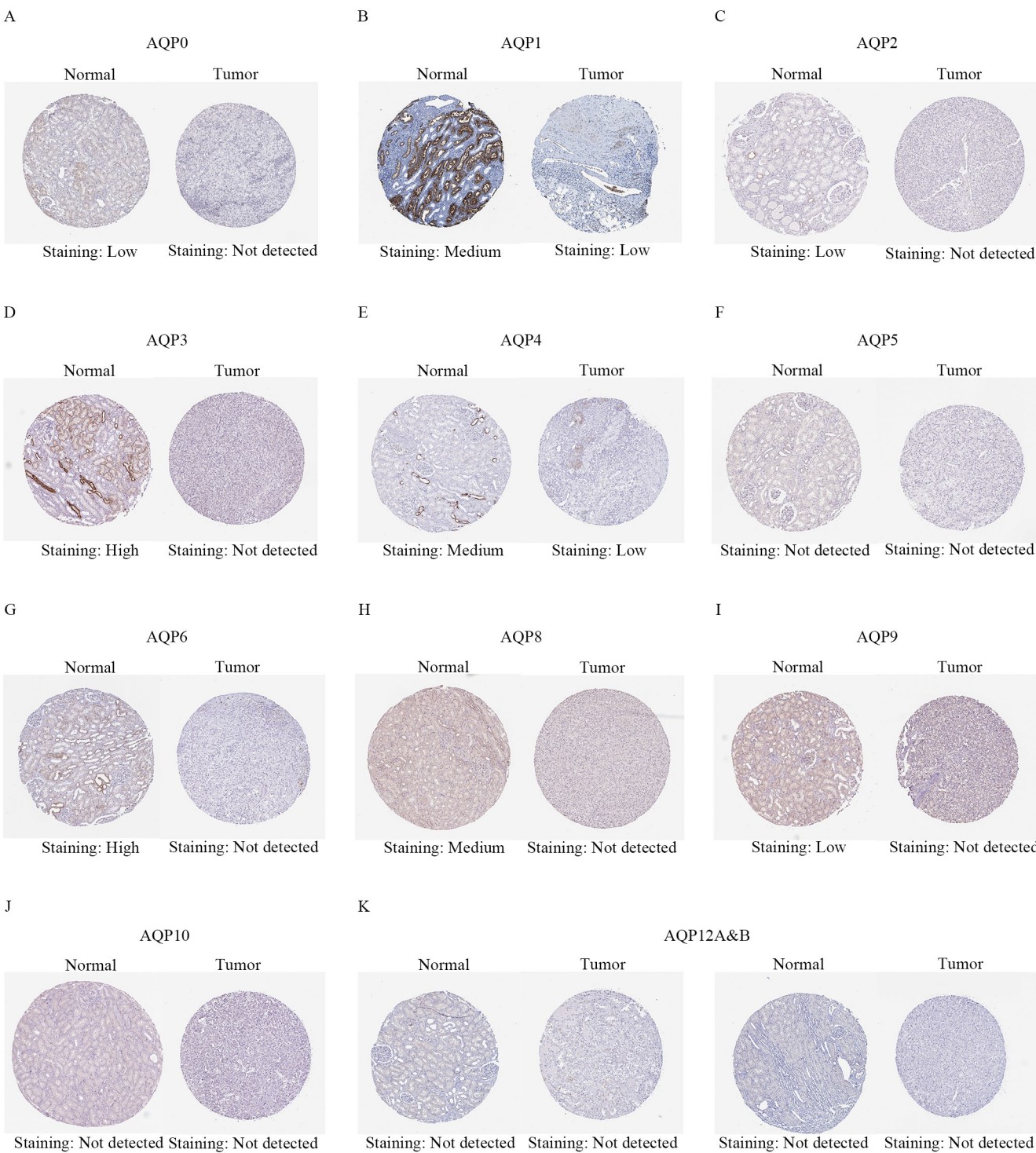

**Fig 3. Representative immunohistochemistry images of AQPs in ccRCC and normal kidney tissues (Human Protein Atlas). (A, C, D, G–I)** AQP0/2/3/6/8/9 proteins were not expressed in ccRCC tissues, whereas their low, medium and high expressions were observed in normal kidney tissues. **(B, E)** Low protein expressions of AQP1/4 were found in ccRCC tissues, while their medium protein expressions were observed in normal kidney tissues. **(F, J, K)** AQP5/10/12A&B proteins were expressed neither in ccRCC nor normal kidney tissues. Republished from https://www.proteinatlas.org/ under a CC BY license, with permission from inger åhlén, original copyright 2021.

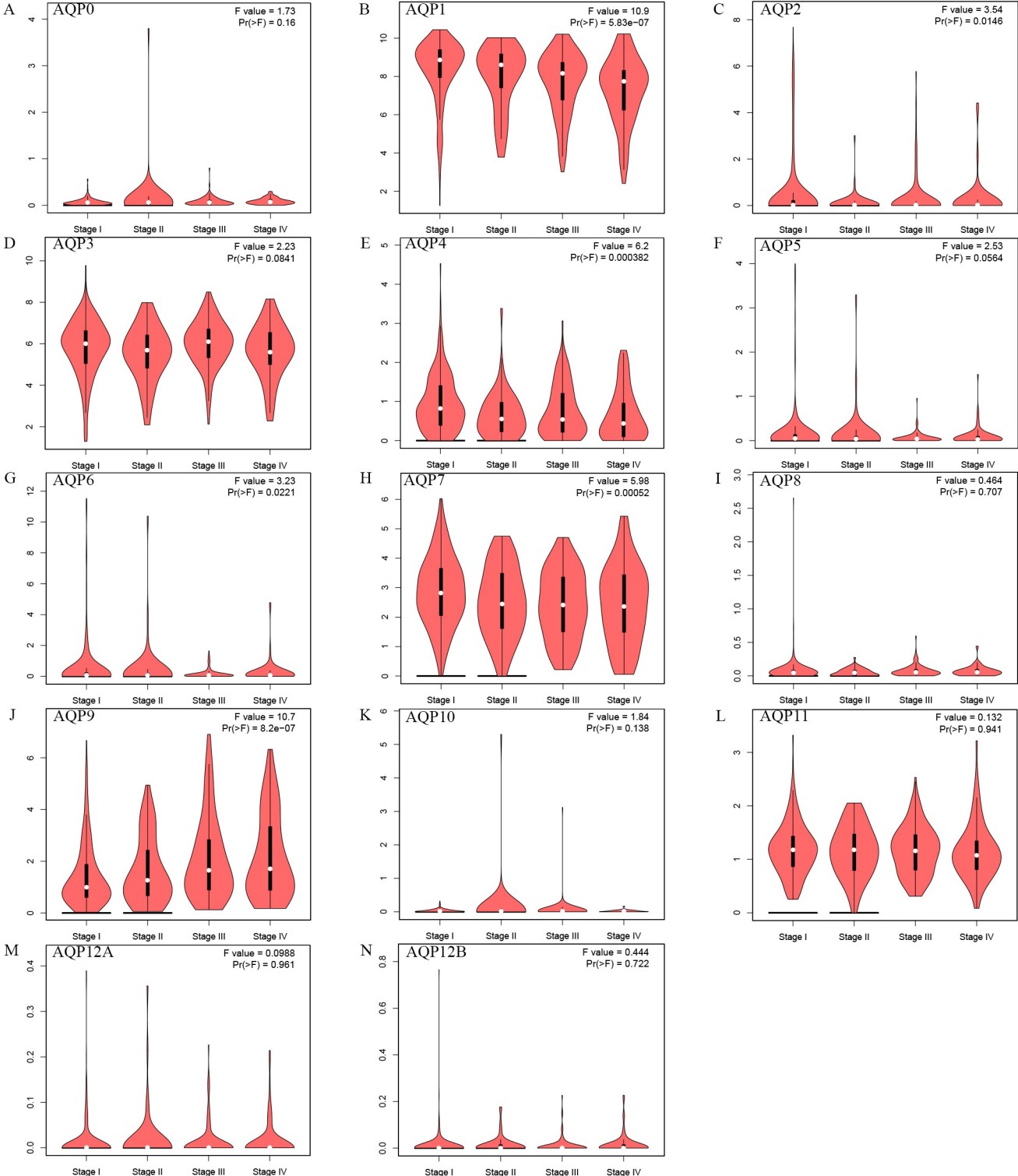

**Fig 4. Relationship between mRNA expression of AQPs and individual cancer stages of ccRCC patients (GEPIA).** The figure was visualized using violin plots. mRNA expression levels of AQP1/2/4/6/7/9 were remarkably correlated with patients' individual cancer stages, while the expression across stages did not significantly differ for AQP0/3/5/8/10/11/12A/12B. The method for differential expression gene analysis was one-way ANOVA, using the pathological stage as the variable for calculating differential expression. The F-value was the test statistic from the F-test. The Pr (>F) was the p-value of the F-test and the p-value < 0.05 was significant.

AQP6 (0.0221), AQP7 (0.00052), AQP9 (8.2e−07) had significant correlations with the pathological stages of the tumor. However, the expression across stages did not significantly differ for AQP0/3/5/8/10/11/12. With the progress of the tumor stage, the expression levels of AQP1/2/4/6/7/9 gradually increased. These results indicated that these genes had a certain relationship with tumor progression.

## Genetic alteration in the frequency of AQPs in patients with ccRCC

Based on the TCGA dataset with the maximum sample size, we used the cBioPortal database to conduct a comprehensive analysis of the genetic mutation frequency of AQPs in ccRCC patients. Firstly, an overall analysis of the dataset was conducted (Fig 5A). In all samples, a total of 148 (33.18%) ccRCC patients showed significant changes in AQPs, of which 8 patients had genetic mutations, 6 patients had gene amplification, 5 patients had deep gene deletions, 119 patients had gene transcription up-regulation, and 10 patients had polygenic mutations. In ccRCC patients, the genetic mutation rates of AQPs were 0%, 5%, 2.5%, 1.8%, 3%, 2.9%, 0.7%, 6%, 0.9%, 4%, 1.1%, 2.7%, 6% and 7%. Enhanced mRNA expression is the most common change in these samples (Fig 5B). Next, the co-expression relationship of AQPs was analyzed (Fig 5C). There were moderate or high positive correlations among the following AQPs: AQP0 and AQP8, AQP1 and AQP7, AQP2/6 and AQP5, AQP8 and AQP9/11, AQP12A and 12B. There was a moderate negative correlation between AQP1 and AQP9. In addition, the relationships between gene expression mutations of AQPs and the survival prognosis of ccRCC patients were explored. Kaplan-Meier curve was used to reveal the OS and DFS in ccRCC patients with mutations or non-mutations (Fig 5D and 5E). Survival results indicated that there was no noticeable discrepancy existed between the altered group and the unaltered group.

## Co-expression and functional enrichment analysis of AQPs and neighbor genes in ccRCC patients

Firstly, the PPI between AQPs was analyzed and constructed through STRING to clarify the potential interactive relationship. The result showed that the network graph contained a total of 14 nodes and 54 edges (PPI enrichment p-value <1.0e-16). The main biological processes of these differential expression levels of AQPs were associated with one-carbon compound transport, renal water transport, water transport, cellular response to mercury ion, and glycerol transport (Fig 6A).

Secondly, a network of AQPs and functionally related genes was further constructed to explore the underlying mechanism of AQPs in ccRCC through GeneMANIA. The result showed that 20 genes were closely related to the regulation and function of AQPs, namely RP11-407P15.2, FP325317.1, RP5-877J2.1, FAM188B, FAM188A, GJA8, GJA3, OR10H3, EXOC4, CRYAA, DMD, EXOC3, STX4, GJA1, GJA4, GJB2, GJA5, GJD2, RP5-864K19.6, and GJC3. The biological processes of these genes were mainly enriched in cell communication by chemical coupling, glycerol transport, atrial ventricular junction remodeling, positive regulation of cell communication by chemical coupling and cellular response to mercury ion (Fig 6B).

Then Metascape was used to analyze the biological functions of AQPs and the aforementioned co-expressed genes. The results showed the 9 most enriched terms, including passive transport by aquaporins, gap junction assembly, gap junction-mediated intercellular transport, water homeostasis, pid e-cadherin stabilization pathway, protein homotetramerization, carbon dioxide transport, bile secretion, and response to elevated platelet cytosolic $Ca^{2+}$. In addition, we had constructed networks of enriched terms that were colored based on the cluster and p-

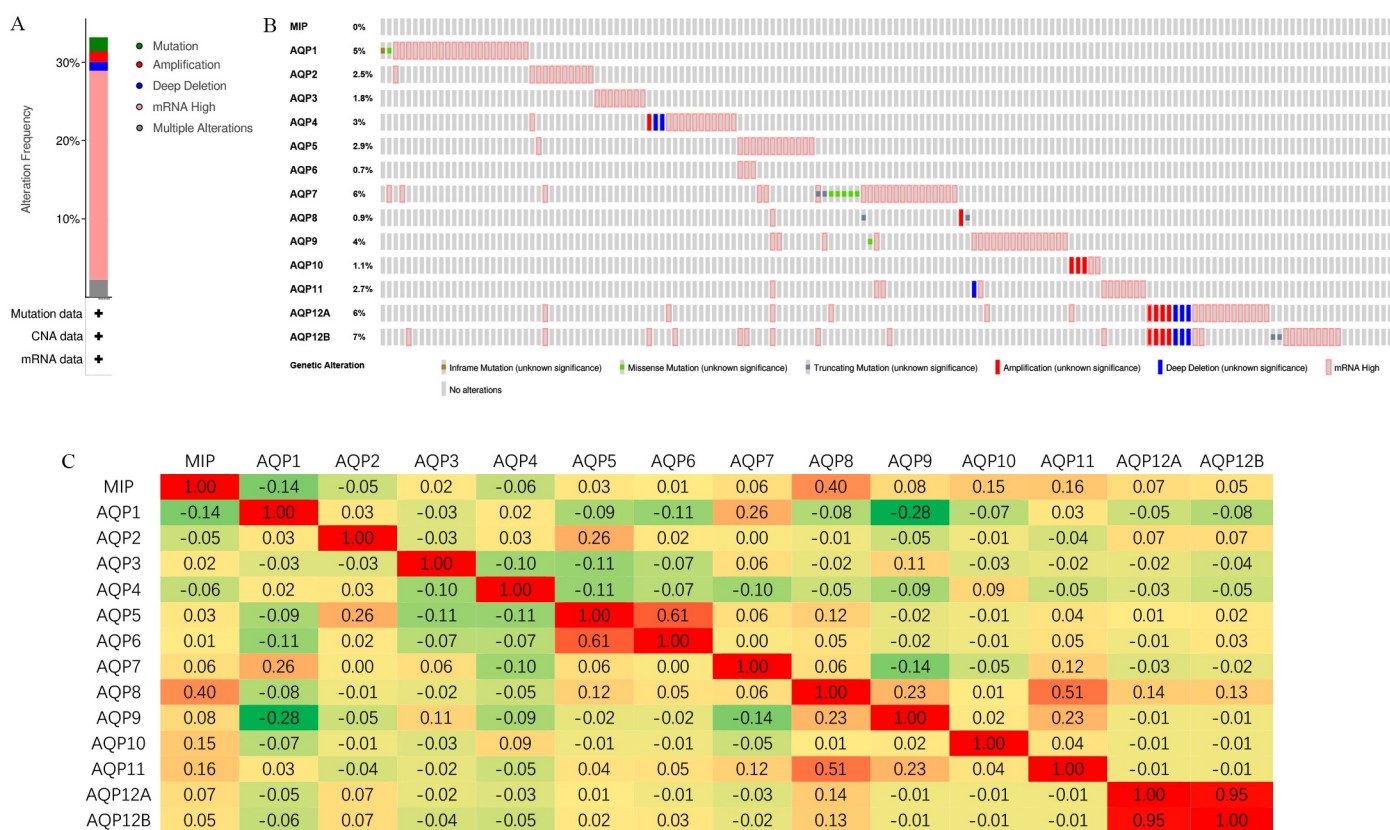

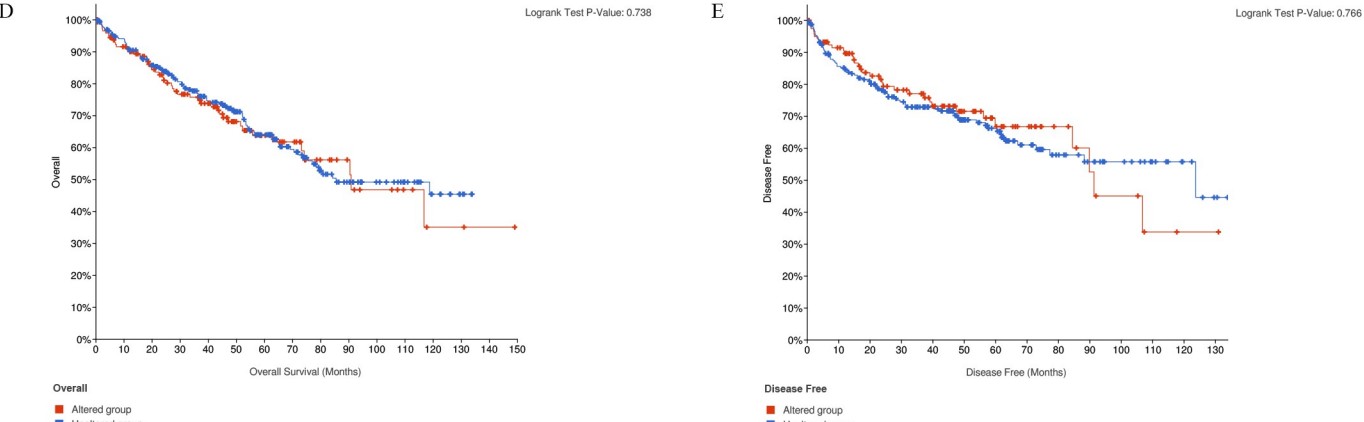

**Fig 5. Genetic mutations in AQPs and their association with the OS and DFS of ccRCC patients (cBioPortal).** (**A**) The bar plot showed the overall alterations of AQPs in the ccRCC samples. The vertical axis represented the percentage of alterations, and the horizontal axis represented the selected tumor sample dataset (ccRCC). Among them, the green indicated mutation, the red was amplification, the blue was deep deletion, the pink was high mRNA expression, and the gray was multiple alterations. A high mutation rate (33.18%) could be observed in ccRCC patients. (**B**) Based on a query of AQPs, cBioPortal summarized genomic alterations in all queried genes across the ccRCC sample set. Each row represented a gene, and each column represented a tumor sample. Brown squares indicated inframe mutations, green squares were missense mutations, gray squares were truncating mutations, red bars were gene amplifications, blue bars were deep deletions, and gray bars with red borders were high mRNA expressions. Mutation rates of each gene were 0%, 5%, 2.5%, 1.8%, 3%, 2.9%, 0.7%, 6%, 0.9%, 4%, 1.1%, 2.7%, 6% and 7%, respectively. (**C**) The figure showed the correlation between AQPs in ccRCC. For each queried gene, Pearson's correlation coefficients were calculated and displayed in the corresponding boxes. (**D, E**) The figure indicated the survival difference between altered and unaltered groups. Genetic alterations of AQPs were not associated with the OS or DFS of ccRCC patients.

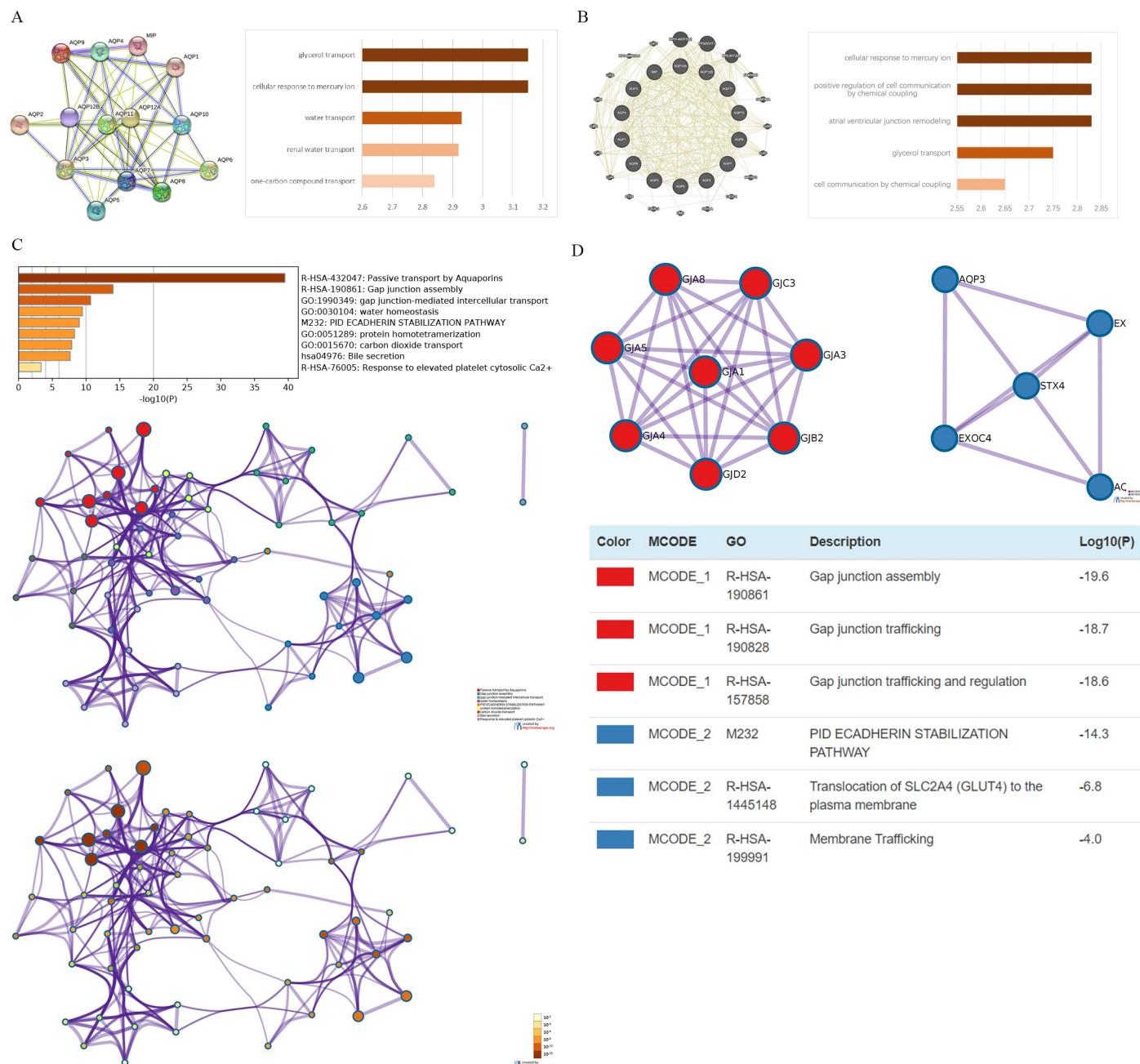

**Fig 6. The enrichment analysis of AQPs and their similar genes in ccRCC (STRING GeneMANIA Metascape).** (**A**) The PPI network and biological process (BP) enrichment analysis of AQPs were constructed through STRING (top5). (**B**) The PPI network and BP enrichment analysis of AQPs and 20 most functionally related genes were constructed through GeneMANIA (top5). (**C**) The gene ontology (GO) enrichment analysis of AQPs and 20 most functionally related neighboring genes in ccRCC (p< 0.05), each GO term is colored by cluster and p-values. (**D**) The two most significant mCODE components and independent functional enrichment analysis were identified through AQPs and 20 most functionally related neighboring genes.

value (Fig 6C). To further analyze the connection between ccRCC and AQPs, we constructed a PPI network and performed mCODE analyses. Then the most significant module information was extracted, and the results suggested that the biological processes were mainly enriched in gap junction assembly, gap junction trafficking, gap junction trafficking and regulation, pid

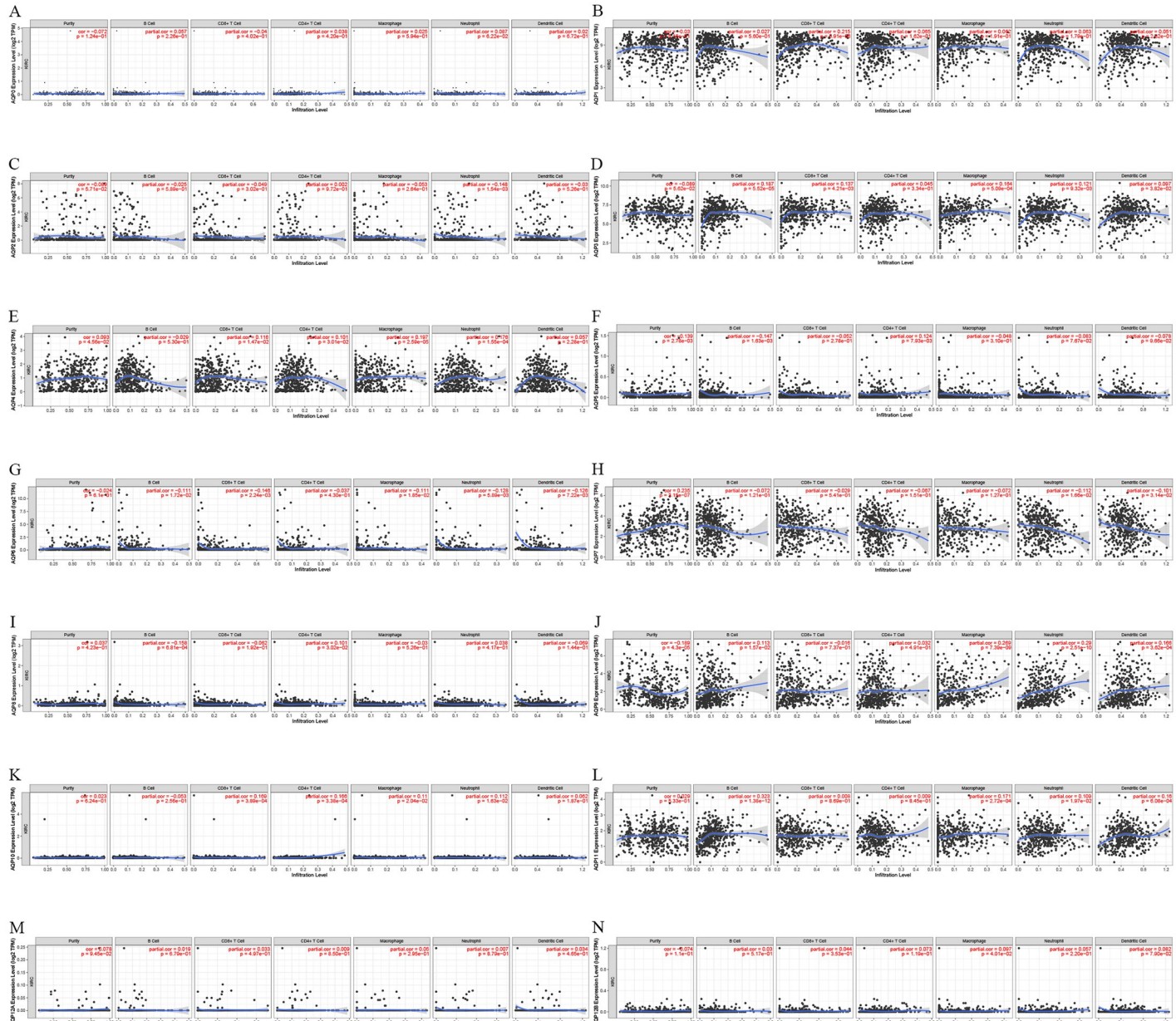

**Fig 7. Correlations of B cells, CD8+T cells, CD4+T cells, macrophages, neutrophils, dendritic cells and AQPs in ccRCC (TIMER).** Correlations of mRNA expression levels between cell markers of B cells, CD8+T cells, CD4+T cells, macrophages, neutrophils, dendritic cells and AQPs in ccRCC patients were shown. The 'partial cor' indicated the purity-corrected partial Spearman's rho value, and the 'p' referred to 'p-value' (p-value < 0.05 was significant).

ecadherin stabilization pathway, translocation of SLC2A4 to the plasma membrane and membrane trafficking (Fig 6D).

Overall, AQPs played a significant role in the assembly, trafficking and regulation of gap junctions between cells. Through this special intercellular connection, AQPs regulate the intercellular transport of water, glycerol, carbon dioxide and other substances, thereby maintaining the normal physiological state of cells.

## Immune cell infiltration of AQPs in patients with ccRCC

Considering that inflammation and infiltrating immune cells would affect the prognosis of ccRCC, we used the TIMER database to comprehensively study the correlation between differentially expressed AQPs and the infiltration of immune cells (Fig 7 and Table 2). It could be found that not every gene of AQPs was closely related to the immune cell infiltration. Individuals with significant immune infiltration correlations were AQP3/4/6/9/10/11. There were also differences in the expression levels of mRNA and protein of these genes in ccRCC, which might mean that there was a certain relationship among the abnormal expression levels of AQPs and the resulting inflammatory microenvironment as well as tumor progression. At the same time, we could observe that AQP3/4/9/10/11 were mainly positively correlated with each immune cell, while AQP6 was negatively correlated with the infiltration degree of immune cells, which suggested that the effects of different AQPs in ccRCC might not be completely consistent. Next, it could be noticed that each gene of AQPs was primarily associated with the infiltration of B cells, macrophages and neutrophils, which indicated that these three immune cells might played a major role in the progression of ccRCC.

## Prognostic value of AQPs in ccRCC

To evaluate whether the different mRNA expression levels of AQPs were related to the survival and prognosis of ccRCC patients, GEPIA was used to analyze AQPs individually. Results revealed that high mRNA expression levels of AQP0($P = 0.013$), AQP8($P = 0.024$), AQP9 ($P = 8.3E-04$) were significantly related to shorter OS of patients with ccRCC while high mRNA expression levels of AQP1($P = 1.8E-09$), AQP4($P = 1.1E-04$), AQP7($P = 0.011$) were associated with longer OS time (Fig 8). The results also demonstrated that high expression of AQP1($P = 4.2E-08$) and AQP7($P = 5.2E-03$) were associated with better prognosis as well as high AQP9($P = 2.1E-03$) expression was correlated with a poorer prognosis of the DFS in ccRCC. AQP12A/12B lacks relevant data temporarily (Fig 9).

After mRNA expression levels of AQP0/1/4/7/8/9 were found to be significantly related to the survival of ccRCC patients, we tried to explore the independent prognostic value of AQPs for OS in ccRCC patients. The clinical data of 526 patients were downloaded and compiled through the TCGA database (Table 3). Then we conducted the univariate and multivariate Cox regression analysis based on these. We found that advanced age (HR = 1.735, 95%CI: 1.276–2.358, and $p<0.001$), high-grade AJCC-T staging (HR = 3.139, 95%CI: 2.316–4.254, and $p<0.001$), lymph node metastasis (HR = 3.423, 95%CI: 1.816–6.451, and $p<0.001$), distant metastasis (HR = 4291, 95%CI: 3.139–5.865, and $p<0.001$), high pathologic stage (HR = 3.832, 95%CI: 2.789–5.266, and $p<0.001$), high histologic grade (HR = 2.629, 95%CI: 1.866–3.704, and $p<0.001$), high mRNA expression of AQP0 (HR = 1.778, 95%CI: 1.292–2.446, and $p<0.001$), AQP9 (HR = 1.847, 95%CI: 1.356–2.518, and $p<0.001$) and low mRNA expression of AQP1 (HR = 0.418, 95%CI: 0.305–0.573, and $p< 0.001$), AQP4 (HR = 0.631, 95%CI: 0.466–0.854, and $p = 0.003$) were related to shorter OS of ccRCC patients (Table 4). The multivariate Cox regression analysis showed that low mRNA expression of AQP1 (HR = 0.484, 95%CI: 0.308–0.759, and $p = 0.002$) and long mRNA expression of AQP9 (HR = 1.622, 95%CI: 1.032–2.550, and $p = 0.036$) were independently associated with significantly shorter OS of ccRCC patients (Tables 5–8).

## Discussion

As a functional protein that exists widely in the cell membranes of various cells, AQPs are assembled by four unique water channel monomers. The transmembrane transport of water,

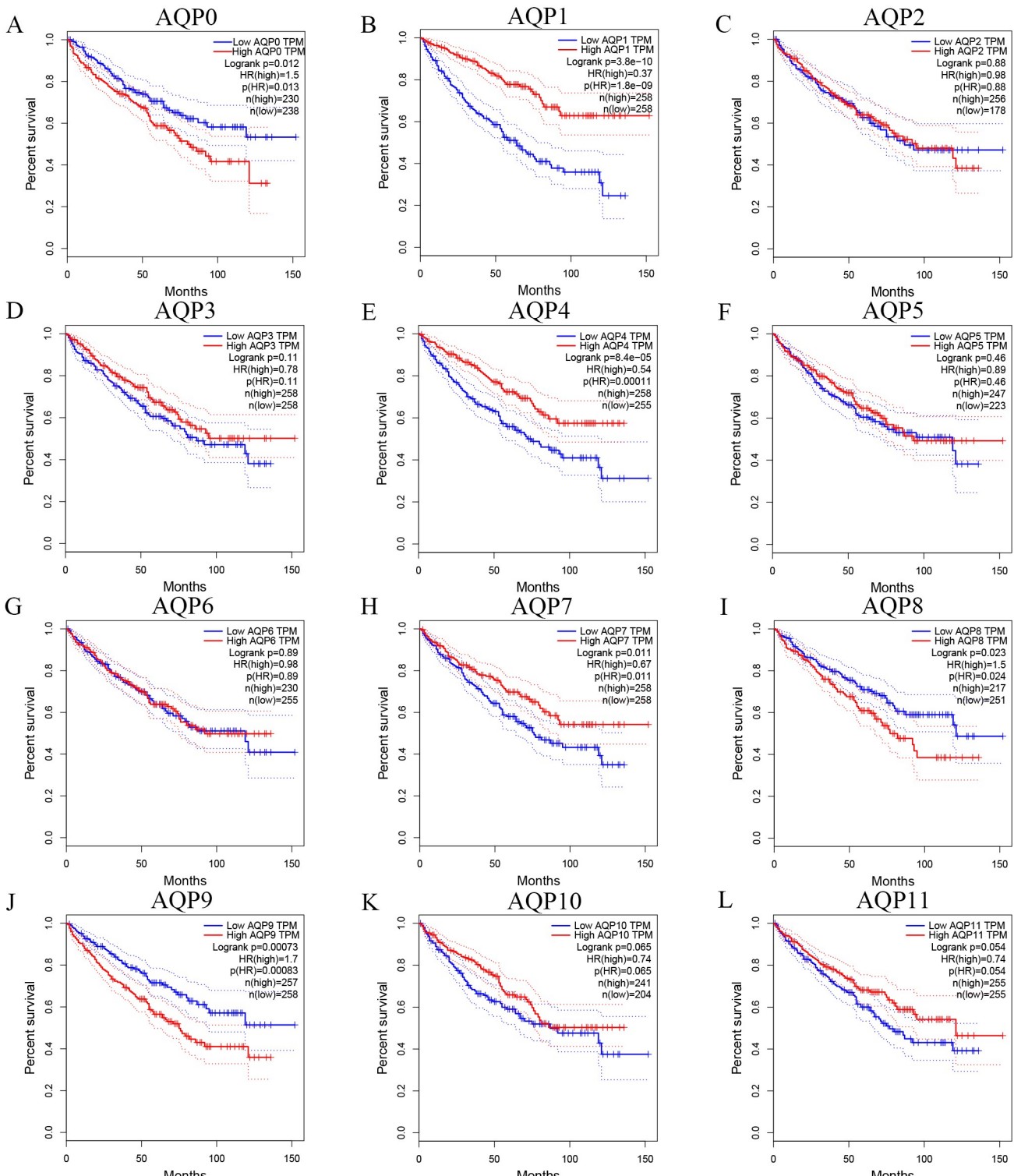

**Fig 8. Prognostic significance of AQPs for the OS of ccRCC patients (GEPIA).** High mRNA expression levels of AQP0/8/9 were significantly related to shorter OS of patients with ccRCC while high mRNA expression levels of AQP1/4/7 were obviously associated with longer OS time.

**Table 2. The correlations of AQPs expression with immune infiltration level in ccRCC.**

| genes | B cells | CD8+T cells | CD4+T cells | macrophages | neutrophils | dendritic cells |
|---|---|---|---|---|---|---|
| AQP0 | 0.057 | -0.04 | 0.038 | 0.025 | 0.087 | 0.02 |
| AQP1 | 0.027 | **0.215** | 0.065 | 0.062 | 0.063 | 0.051 |
| AQP2 | -0.025 | -0.049 | 0.002 | -0.053 | **-0.148** | -0.03 |
| AQP3 | **0.187** | **0.137** | 0.045 | **0.164** | **0.121** | **0.097** |
| AQP4 | -0.029 | **0.116** | **0.101** | **0.197** | **0.176** | 0.057 |
| AQP5 | **-0.147** | -0.052 | **0.124** | -0.048 | -0.083 | -0.078 |
| AQP6 | **-0.111** | **-0.146** | -0.037 | **-0.111** | **-0.128** | **-0.126** |
| AQP7 | -0.072 | -0.029 | -0.067 | -0.072 | **-0.112** | **-0.101** |
| AQP8 | **-0.158** | -0.062 | **0.101** | -0.03 | 0.038 | -0.069 |
| AQP9 | **0.113** | -0.016 | 0.032 | **0.269** | **0.29** | **0.166** |
| AQP10 | -0.053 | **0.169** | **0.166** | **0.11** | **0.112** | 0.062 |
| AQP11 | **0.323** | 0.008 | 0.009 | **0.171** | **0.109** | **0.16** |
| AQP12A | 0.019 | 0.033 | 0.009 | 0.05 | 0.007 | 0.034 |
| AQP12B | 0.03 | 0.044 | 0.073 | **0.097** | 0.057 | 0.082 |

The table showed the associations between individual AQP genes and different immune infiltration by listing the corrected spearman correlation coefficient. The bold values indicated statistical significance (p-value < 0.05).

gas, glycerin, ions and other substances is completed through the channels which maintain the normal physiological morphology and biological functions of various cells [29].

At present, many studies have shown that under normal circumstances, AQPs can perform physiological functions in different parts of the kidney to maintain urine osmotic pressure and cell homeostasis. However, different kidney diseases, such as nephrogenic diabetes insipidus (NDI), autosomal dominant polycystic kidney disease (ADPKD), and acute kidney injury (AKI), would be caused by the occurrence of mutations or deletions of AQPs. Nevertheless, it is different from other diseases related to AQPs, for instance, the high expression of AQPs will aggravate the edema of astrocytoma and eventually lead to an increase in mortality; the severity of chronic obstructive pulmonary disease (COPD) and asthma is related to the high expression of AQPs [9, 30, 31].

At the same time, AQPs are involved in the process of cell migration, cell proliferation, and cell adhesion which are closely related to the occurrence and development of tumor cells [32]. At present, a large number of studies have analyzed the potential relationship between AQP1 and RCC. The conclusion is that the titer of AQP1 in urine can be used as a target indicator for early diagnosis of RCC patients and a sensitive monitoring indicator for RCC patients after surgery. The disadvantage is that these experiments measure the protein level of AQP1 in urine instead of the mRNA expression level, and also face the problem of insufficient sample size. Meanwhile, there is a lack of research on other members of AQPs, which prompts us to conduct this systematic analysis to reveal the expression of AQPs in ccRCC as much as possible [33–35].

AQP0 is a channel protein that also has the function of regulating cell adhesion. With the deepening of research, the function of AQP0 to transport water and gas has gradually been recognized [36]. At the same time, Karin L et al. and Kalman et al. found that pH and calcium ion concentration were related to the expression of AQP0, but the specific pathway and molecular mechanism were still unclear [37, 38]. In our research on AQP0, we found that the mRNA expression level of AQP0 in ccRCC was relatively higher than that of normal kidney tissue, but there was no significant statistical difference. The mutual verification results of multiple

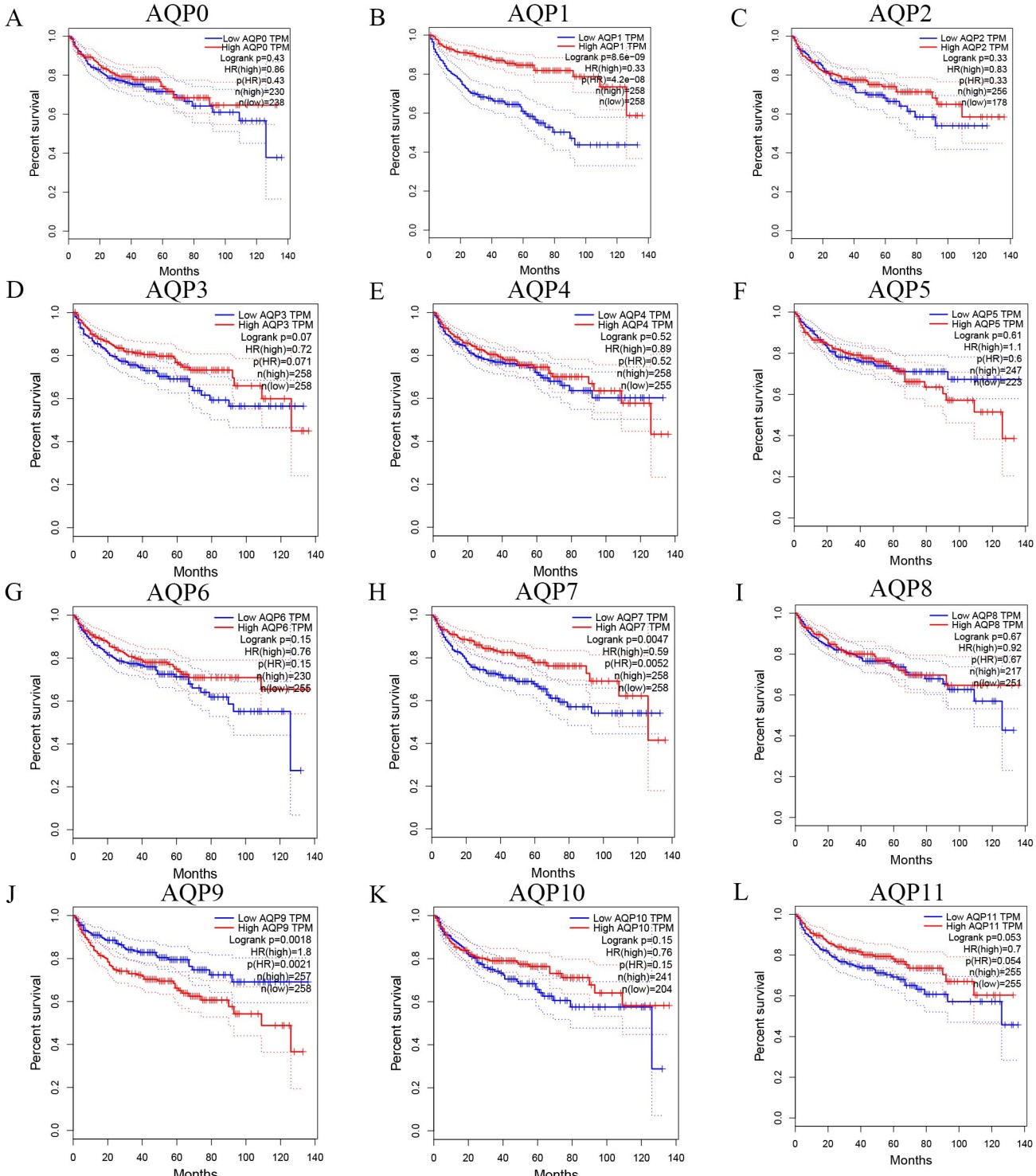

**Fig 9. Prognostic significance of AQPs for the DFS of ccRCC patients (GEPIA).** High expressions of AQP1/7 were associated with a better prognosis while high AQP9 expression was correlated with a poorer prognosis of the DFS in ccRCC patients.

**Table 3. Basic characteristics of 526 ccRCC patients.**

| Variables | ccRCC patients (N = 526) |
|---|---|
| **Age (Mean±SD)** | 60.6±12.1 |
| **Follow-up [Days, median (IQR)]** | 1200 (526–1928) |
| **Gender** | |
| Male | 342 (65%) |
| Female | 184 (35%) |
| **Status** | |
| Alive | 353 (67.1%) |
| Dead | 173 (32.9%) |
| **Race** | |
| Asian | 8 (1.5%) |
| Black or african american | 56 (10.6%) |
| White | 455 (86.5%) |
| Unknown | 7 (1.3%) |
| **AJCC-T** | |
| T1 | 269 (51.1%) |
| T2 | 68 (12.9%) |
| T3 | 178 (33.8%) |
| T4 | 11 (2.1%) |
| **AJCC-N** | |
| N0 | 239 (45.4%) |
| N1 | 16 (3.0%) |
| NX | 271 (51.5%) |
| **AJCC-M** | |
| M0 | 416 (79.1%) |
| M1 | 78 (14.8%) |
| MX | 30 (5.7%) |
| Unknown | 2 (0.4%) |
| **Pathologic stage** | |
| I | 263 (50.0%) |
| II | 56 (10.6%) |
| III | 122 (23.2%) |
| IV | 82 (15.6%) |
| Unknown | 3 (0.6%) |
| **Histologic grade** | |
| G1 | 14 (2.7%) |
| G2 | 224 (42.6%) |
| G3 | 205 (39.0%) |
| G4 | 75 (14.3%) |
| GX | 5 (1.0%) |
| Unknown | 3 (0.6%) |
| **Laterality** | |
| Left | 248 (47.1%) |
| Right | 277 (52.7%) |
| Bilateral | 1 (0.2%) |
| **Treatment type** | |
| Radiation therapy | 260 (49.4%) |
| Pharmaceutical therapy | 266 (50.6%) |

**Table 4. Univariate analysis of overall survival in 526 ccRCC patients.**

| Variable | Univariate | |
|---|---|---|
| | HR (95%CI) | p-value |
| **Age** | | |
| ≤60 | Reference | |
| >60 | 1.735 (1.276–2.358) | <0.001* |
| **Gender** | | |
| Male | Reference | |
| Female | 1.050 (0.770–1.432) | 0.757 |
| **AJCC-T** | | |
| T1-T2 | Reference | |
| T3-T4 | 3.139 (2.316–4.254) | <0.001* |
| **AJCC-N** | | |
| N0 | Reference | |
| N1 | 3.423 (1.816–6.451) | <0.001* |
| **AJCC-M** | | |
| M0 | Reference | |
| M1 | 4.291 (3.139–5.865) | <0.001* |
| **Pathologic stage** | | |
| I-II | Reference | |
| III-IV | 3.832 (2.789–5.266) | <0.001* |
| **Histologic grade** | | |
| G1-G2 | Reference | |
| G3-G4 | 2.629 (1.866–3.704) | <0.001* |
| **Gene expression** | | |
| AQP0 | 1.778 (1.292–2.446) | <0.001* |
| AQP1 | 0.418 (0.305–0.573) | <0.001* |
| AQP2 | 1.046 (0.773–1.414) | 0.772 |
| AQP3 | 0.758 (0.562–1.023) | 0.070 |
| AQP4 | 0.631 (0.466–0.854) | 0.003* |
| AQP5 | 0.965 (0.716–1.301) | 0.817 |
| AQP6 | 1.055 (0.781–1.427) | 0.726 |
| AQP7 | 0.758 (0.561–1.024) | 0.071 |
| AQP8 | 1.673 (1.231–2.275) | 1.673 |
| AQP9 | 1.847 (1.356–2.518) | <0.001* |
| AQP10 | 1.022 (0.751–1.391) | 0.889 |
| AQP11 | 0.875 (0.649–1.180) | 0.383 |
| AQP12A | 1.000 (0.589–1.699) | 0.999 |
| AQP12B | 1.409 (0.908–2.186) | 0.126 |

datasets were more consistent. At the same time, the mutation rate of AQP0 in ccRCC was almost zero, and there was no obvious relationship with tumor grade. However, AQP0 showed differences in survival analysis, and high expression was associated with worse OS in patients. At present, the research on AQP0 mainly focuses on the lens, and there are relatively few studies in other tumors. However, the function of AQP0 to achieve cell adhesion through the C-loop extracellular domain may be a potential factor to promote tumor cell migration, metastasis and colonization [39].

**Table 5. Multivariate analysis of overall survival in 526 ccRCC patients.**

| Variable | Multivariate | |
|---|---|---|
| | HR (95%CI) | p-value |
| **Age** | | |
| ≤60 | Reference | |
| >60 | 1.810 (1.153–2.840) | 0.01 |
| **AJCC-T** | | |
| T1-T2 | Reference | |
| T3-T4 | 1.889 (1.184–3.014) | 0.008 |
| **AJCC-N** | | |
| N0 | | |
| N1 | | |
| **AJCC-M** | | |
| M0 | Reference | |
| M1 | 3.622 (2.229–5.883) | <0.001 |
| **Pathologic stage** | | |
| I-II | | |
| III-IV | | |
| **Histologic grade** | | |
| G1-G2 | Reference | |
| G3-G4 | 1.794 (1.089–2.957) | 0.022 |
| **Gene expression** | | |
| AQP0 | | |

**Table 6. Multivariate analysis of overall survival in 526 ccRCC patients.**

| Variable | Multivariate | |
|---|---|---|
| | HR (95%CI) | p-value |
| **Age** | | |
| ≤60 | Reference | |
| >60 | 1.638 (1.067–2.517) | 0.024 |
| **AJCC-T** | | |
| T1-T2 | Reference | |
| T3-T4 | 2.013 (1.289–3.143) | 0.002 |
| **AJCC-N** | | |
| N0 | | |
| N1 | | |
| **AJCC-M** | | |
| M0 | Reference | |
| M1 | 3.049 (1.916–4.852) | <0.001 |
| **Pathologic stage** | | |
| I-II | | |
| III-IV | | |
| **Histologic grade** | | |
| G1-G2 | | |
| G3-G4 | | |
| **Gene expression** | | |
| AQP1 | 0.484 (0.308–0.759) | 0.002 |

**Table 7. Multivariate analysis of overall survival in 526 ccRCC patients.**

| Variable | Multivariate | |
|---|---|---|
| | HR (95%CI) | p-value |
| **Age** | | |
| ≤60 | Reference | |
| >60 | 1.670 (1.087–2.565) | 0.019 |
| **AJCC-T** | | |
| T1-T2 | Reference | |
| T3-T4 | 1.931 (1.220–3.056) | 0.005 |
| **AJCC-N** | | |
| N0 | | |
| N1 | | |
| **AJCC-M** | | |
| M0 | Reference | |
| M1 | 3.050 (1.912–4.867) | <0.001 |
| **Pathologic stage** | | |
| I-II | | |
| III-IV | | |
| **Histologic grade** | | |
| G1-G2 | Reference | |
| G3-G4 | 1.728 (1.058–2.821) | 0.029 |
| **Gene expression** | | |
| AQP4 | | |

**Table 8. Multivariate analysis of overall survival in 526 ccRCC patients.**

| Variable | Multivariate | |
|---|---|---|
| | HR (95%CI) | p-value |
| **Age** | | |
| ≤60 | Reference | |
| >60 | 1.673 (1.084–2.580) | 0.02 |
| **AJCC-T** | | |
| T1-T2 | Reference | |
| T3-T4 | 1.798 (1.120–2.887) | 0.015 |
| **AJCC-N** | | |
| N0 | | |
| N1 | | |
| **AJCC-M** | | |
| M0 | Reference | |
| M1 | 3.077 (1.922–4.926) | <0.001 |
| **Pathologic stage** | | |
| I-II | | |
| III-IV | | |
| **Histologic grade** | | |
| G1-G2 | Reference | |
| G3-G4 | 1.653 (1.002–2.729) | 0.049 |
| **Gene expression** | | |
| AQP9 | 1.622 (1.032–2.550) | 0.036 |

AQP1 plays an important role in kidney cancer. AQP1 is abundantly expressed in the proximal tubule, descending thin limbs of Henle and vasa recta, and it plays a key role in the concentration of urine under physiological conditions [40, 41]. As mentioned above, after the discovery of AQP1, successive studies had explored the role of AQP1 in kidney disease, and finally found that the protein level of AQP1 in urine can be used as a diagnostic indicator for early-stage ccRCC. We have found through research that the mRNA and protein expression levels of AQP1 in ccRCC were significantly lower than those in normal kidney tissues, and the mRNA expression level of AQP1 gradually decreased with the increase of tumor grade. These were in line with the results of Ying Huang et al. [33]. In terms of biological processes, AQP1 played a role in one-carbon compound transport, renal water transport, water transport, cellular response to mercury ion and glycerol transport. At the same time, the increased expression of AQP1 was significantly related to better OS and DFS in ccRCC patients, which might indicate that AQP1 was a promising tumor suppressor gene for ccRCC patients. David TV et al. showed that the expression level of AQP1 in ccRCC was significantly lower than that in normal kidney tissue by Western blot analyses, and the results of Ying Huang et al. showed that the group with high AQP1 expression level had a better prognosis. The experimental results of the two can increase the credibility of our results undoubtedly [33, 42].

AQP2 is a channel protein mainly expressed in the collecting duct. Under normal physiological conditions, it is regulated by vasopressin and plays an important function in concentrating urine through turning inside and outside the cell membrane [43]. Accompanied by abnormal expression of AQP2, patients will develop nephrogenic diabetes insipidus, which further confirms the physiological function of AQP2 [44]. We explored through different databases and found that AQP2 expression was found in the gene-sequencing results of Gumz et al. [24], and their results were consistent. The expression level of AQP2 in ccRCC was significantly lower than that in normal tissues. The results provided by the TCGA database showed that although the expression of AQP2 was low in ccRCC, the number of samples that could be detected AQP2 expression is small. At the same time, although the mRNA expression level of AQP2 was related to tumor grade, it had no significant statistical significance with the patient's OS and DFS. Liao SF et al. demonstrated that the mRNA and protein expression levels of AQP2 were significantly reduced in ccRCC, which verified the accuracy of our results to a certain extent [45].

AQP3 and AQP4 are two members mainly expressed in the outer membrane of the base of the collecting duct [31]. In addition to transporting water out of the cytoplasm, AQP3 and AQP4 can also promote the transmembrane transport of glycerol and hydrogen peroxide, thereby regulating a series of intracellular signal transduction and affecting cell functions, such as cells proliferation, apoptosis, and migration [31, 40]. At the same time, studies have shown that AQP3-knockout mice had obvious dysfunction of urine concentration. When AQP3 and AQP4 were knocked out at the same time, the symptoms would be more obvious. But when only AQP4 was knocked out, no changes in the expression of other members of AQPs were found, which meant that no compensatory effect had occurred [41]. In the results we obtained, Higgins et al. detected AQP3 mRNA expression in the tissues of ccRCC patients [28]. The results displayed by TCGA were more adequate than those of AQP2. The mRNA expression levels of AQP3 and AQP4 in ccRCC patients were significantly lower than those in normal kidney tissues. At the same time, the protein expression levels of AQP3 and AQP4 in ccRCC were significantly lower than those in normal tissues, which indicated that the functions of AQP3 and AQP4 were reduced after mutation. However, the difference was that the dysfunction of AQP3 was more obvious than that of AQP4, which further explained the association between the lack of AQP3 and nephrogenic urinary incontinence. The biological process of AQP3 and AQP4 participated in water transport and maintenance of water homeostasis in the

kidney, which was related to their similarities in their physiological structures and anatomical locations. Surprisingly, AQP4 was more significantly related to the OS of ccRCC patients than AQP3, and the high mRNA expression level of AQP4 was related to better OS in patients.

AQP5 and AQP6 are also two channel proteins mainly expressed in the collecting duct [31]. AQP5 is a gene related to the occurrence and development of many tumors. It is involved in epithelial-mesenchymal transition, which is an important process of participating in invasion and metastasis for malignant tumors. At the same time, the abnormal expression of AQP5 can regulate the EGFR pathway and promote the proliferation of tumor cells [46]. Compared with concentrated urine and maintaining water balance, the main function of AQP6 is more inclined to regulate acid-base balance [47]. Existing studies had shown that the permeability of AQP6 anion channels was significantly increased under low pH conditions and colocalized with H+-ATPase in intracellular vesicles of the acid-secreting intercalated cells, suggesting a role of AQP6 in promoting urinary acid secretion. AQP6 had been shown to bind calmodulin in a calcium-dependent manner in vitro. This binding might be involved in AQP6 intracellular vesicle localization and physiological function of AQP6 [41]. In our study, the mRNA and protein expression levels of AQP5 and AQP6 in ccRCC patients were significantly lower than those in normal kidney tissues, which were different from the situation in other types of tumors. There were no significant relationships between the abnormal expression of AQP5 or AQP6 and the OS or DFS of ccRCC patients, which might be related to the histological localization of AQP5 and AQP6.

AQP7 is a transmembrane protein that is mainly expressed in proximal tubules like AQP1 [31]. Existing studies had shown that AQP7 played an important role in regulating glycerol metabolism. It could be observed in mice that were knocked out AQP7. With the decrease of AQP7 expression, the level of glycerol in urine was significantly higher than normal [48]. AQP7 affects intracellular energy metabolism by realizing the transmembrane transport of glycerol. The lack of AQP7 is related to the increase of glycerol kinase activity and triglyceride accumulation in adipose tissue, resulting in secondary development of obesity and insulin resistance [49]. We had obtained that the mRNA expression level of AQP7 in ccRCC patients was significantly lower than that in normal tissues, and the mRNA expression level of AQP7 decreased with the increase of the grade of renal cancer. About the biological processes, AQP7 was mainly involved in the transport of water and glycerol, while AQP1 was also involved in these two processes. In terms of immune infiltration, the increase of the expression level of AQP7 was negatively correlated with the degree of infiltration for most immune cells, which might indicate that AQP7 was closely related to the inflammatory microenvironment of ccRCC. At the same time, the high expression of AQP7 had a significant relationship with better OS and DFS in ccRCC patients. In general, in our research, the role of AQP7 in ccRCC was consistent.

AQP8 is a transmembrane protein mainly expressed in the proximal convoluted tubules and collecting ducts in the kidney [41]. Existing studies had shown that AQP8 had no obvious regulatory effect on urine concentration, but played a role in ammonia transport and hydrogen peroxide transport [31, 40, 47]. In the experiment of knocked-out AQP8 mice, there were no significant differences in the urine concentration between the wild-type mice and AQP8 deletion mice [50]. AQP9 plays an important role in the transport of many substances in the kidney, of which the more important function is regulating the transport of glycerol [51, 52]. At the same time, the abnormal expression of AQP9 is closely related to inflammatory cell infiltration. It can regulate a series of inflammation-related signal pathways, such as IL6/JAK-STAT3, IL2-STAT5, and TNFα signaling pathways, in which the IL-6/JAK-STAT3 pathway is abnormally activated in many cancers, and hyperactivity is usually associated with poor clinical prognosis. In the tumor microenvironment, the IL-6/JAK-STAT3 signaling pathway

promotes tumor cell proliferation, invasion, and metastasis, while strongly inhibits tumor immune response [53, 54]. Research on AQP10 in the kidney is relatively scarce. Existing studies showed that the expression of AQP10 in the duodenum and jejunum was more abundant, and was closely related to glycerol transport [52]. In our research results, AQP8/9/10 were more consistently up-regulated in ccRCC patients, and the abnormal expression levels of AQP8/9 were more significant. As for protein expression, the expression levels of AQP8/9/10 in ccRCC patients were down-regulated compared with normal tissues. Among them, AQP9 was more closely related to the clinical prognosis of ccRCC patients. We found that the expression of AQP9 was positively correlated with B cell, macrophage, neutrophil, and dendritic cell, which indicated that AQP9 had a certain relationship with the tumor inflammatory microenvironment. At the same time, the expression of AQP9 was significantly up-regulated with the increase of tumor grade, and the expression of AQP9 was up-regulated. It was also closely related to poor OS and DFS in ccRCC patients. These results indicated that perhaps AQP9 could be used as a diagnostic index and an effective anchor point for targeted therapy in ccRCC patients. Meng XG et al. showed that the mRNA expression level of AQP9 was significantly increased in ccRCC [55]. In addition, JIng JB et al. showed that the protein expression level of AQP9 was significantly increased in ccRCC, the expression level of AQP9 increased with the pathological stage of the tumor. At the same time, AQP9 was associated with the expression of macrophages and the inhibition of the recruitment of NK and CD8+T cells in the tumor microenvironment [56]. These experimental results can further confirm our results.

AQP11 is a transmembrane protein mainly expressed in the endoplasmic reticulum of proximal convoluted tubule cells of the kidney [31]. Under physiological conditions, AQP11 has an effect on transmembrane transport for water, glycerol, urea, and other substances [51]. When the expression deletion of AQP11 occurs, it can lead to the formation of proximal tubule cysts in the kidney caused by PC1 transport defects, which in turn leads to polycystic kidney disease and induces severe renal failure [57, 58]. At the same time, studies had shown that the afunction of AQP11 was associated with acute kidney injury and diabetic nephropathy [59–61]. However, research on the potential relationship between AQP11 and kidney cancer is insufficient. Our research showed that the expression of AQP11 in ccRCC patients was significantly down-regulated, and the high expression of AQP11 was associated with better OS and DFS in ccRCC patients. Because of the high expression of AQP1/7/11 in the proximal convoluted tubules of the kidney, perhaps AQP11 can also become an early diagnostic indicator and treatment target for ccRCC. As a relatively new member of AQPs, AQP12 is considered a super aquaporin [62]. At present, the research on AQP12 is relatively sufficient in the field of pancreatic-related diseases, but the basic research and clinical research related to the kidney are still blank [63]. In our research, the data of AQP12A/12B were relatively inadequate, and some databases were even missing. The physiological functions related to AQP12 and the potential relationship with kidney cancer still need to be further explored.

One thing worth seriously considering is that, as the smallest unit of function, the level of protein plays an important role in the physiological state and the occurrence and development of various diseases. In the research process, we always hope that the expression level of mRNA and protein will show a consistent relationship. Therefore, we used the HPA database to explore the consistency of the protein level and mRNA expression level of AQPs. We found that the two levels of AQP1/2/3/4/5/6 were consistent in normal kidney tissues and ccRCC tissues. But those of AQP0/8/9/10/12 were not consistent. This is because mRNA does not necessarily have a linear relationship with protein expression. There are many levels of regulation of gene expression, and the regulation of transcription level is only one link. Both post-transcriptional regulation and post-translational regulation play a role in the final protein expression [64–68]. At the same time, factors such as mRNA degradation, protein degradation, and

modified folding may cause the mRNA abundance to be inconsistent with the protein expression level [69, 70]. In addition, many genes have translational heterogeneity, and the amount of protein translated will change with time and space [71, 72]. For example, Pannexins is a three-member protein family that forms macroporous ion and metabolite channels in vertebrates [73]. Langlois et al. found that human PANX1 and PANX3 were differentially expressed during skeletal muscle proliferation and differentiation. PANX1 was expressed at low levels in undifferentiated human skeletal myoblasts and muscle cells, and increased with differentiation. However, PANX3 was highly expressed in undifferentiated cells and increased with differentiation [74]. In mouse skin, the level of PANX1 detected in the skin of young mice was higher than that of old skin, while the level of PANX3 remained unchanged [75]. Therefore, in addition to observing the expression level of mRNA, it is also particularly important to confirm the level of protein.

Finally, it is worthy to pay special attention to the independent prognostic factors AQP1 and AQP9 that are finally screened. Through multi-dimensional analysis, it is not difficult to find that there are significant changes in the mRNA expression levels of AQP1 and AQP9 in ccRCC, and both will change significantly with the progression of ccRCC stages. It is worth noting, however, that AQP1 and AQP9 have different trends in these respects. The expression level of AQP1 in ccRCC shows a gradually decreasing trend, while that of AQP9 is the opposite. In terms of mRNA mutation and immune cell infiltration, the expressions of AQP1 and AQP9 are relatively consistent. The mRNA expression levels of AQP1 and AQP9 are positively correlated with the infiltration of most immune cells. Clinical prognosis correlation analysis indicated that the group with a high expression level of AQP1 tends to show better OS and DFS, while the group with a high expression level of AQP9 means poorer OS and DFS. It is well known that the occurrence of most kidney cancers is closely related to the abnormal function of the von Hippel-Lindau (VHL) gene. The abnormality of the VHL gene can lead to the reduction and accumulation of hypoxia-inducible factor (HIF), thereby initiating the transcriptional activation of hypoxia-responsive genes, and ultimately promoting the occurrence and development of cancer [76]. In the pathway AQP1 is involved in, the up-regulation of AQP1 can promote the stabilization of HIF, which in turn makes the high expression group show a better prognosis [33]. Jing JB et al. explored the expression of AQP9 which was consistent with ours, and they also analyzed the relationship between AQP9 and tumor environment. The results showed that in ccRCC, AQP9 promoted tumor-associated macrophage polarization, inhibited the recruitment of natural killer (NK) cells and CD8+ T cells by inhibiting P53 and activating the Janus kinase/signal transducer and activator of transcription (JAK/STAT) pathway, and ultimately stimulated a reorientation of the tumor microenvironment in ccRCC toward tumor-friendly directions [56]. Combining the above analyses of AQP1 and AQP9, we may guess that AQP1 is the tumor suppressor gene of ccRCC while AQP9 is the proto-oncogene of ccRCC. Although both genes belong to the AQPs family, they may play diametrically opposite roles, which are interesting and worthy of further exploration.

Our study provides a comprehensive analysis of AQPs in ccRCC using multiple online databases based on the most popular bioinformatics theories. These available methods are characterized by large sample size, low cost, and the ability to perform large-scale genomics studies and functional analyses. However, our study still has some limitations. First, online databases have limitations. Different databases may yield different results due to differences in the sources of the collected samples. Furthermore, this study only presents the results of bioinformatics analysis based on different online databases. Therefore, further molecular biology experiments, including quantitative real time polymerase chain reaction (qRT-PCR), Western blot, and immunohistochemistry, are required to validate the results of this study.

## Conclusions

In this study, the Oncomine database and the UALCAN database were used to analyze the mRNA of 14 members of AQPs, and the prognostic values of AQPs in ccRCC were analyzed using Kaplan-Meier plotter and GEPIA database. The results showed that in ccRCC patients, the mRNA expression levels of AQP0/8/9/10 were up-regulated in ccRCC, while those of AQP1/2/3/4/5/6/7/11 were down-regulated in ccRCC. The clinical database showed that the high mRNA expression levels of AQP0/8/9 were significantly associated with poor OS. On the contrary, the high levels of AQP1/2/3/4/5/6/7/10/11, especially the high levels of AQP1/4/7 were correlated with better OS in ccRCC patients. Among them, AQP1/7/11 have similarities in the tissue structure and positioning, and they are all highly expressed in the proximal tubules of the kidney. Perhaps AQP7/11 can become a potential diagnostic index and therapeutic target for ccRCC after AQP1. As the only two independent prognostic factors, AQP1 and AQP9 have shown differential expressions in ccRCC and normal kidney tissues as well as significant immune cell infiltration, which indicates that AQP1 and AQP9 may be used as a new prognostic and diagnostic marker in ccRCC. At the same time, each AQPs member may exert its function through different signaling pathways. To further explore the role of AQPs in ccRCC, more refined mechanism research and big data clinical trials are needed. Our study comprehensively analyzed the transcriptomics characteristics and the prognosis value of AQPs in ccRCC. Based on the verification of a large amount of data, the members of AQPs are expected to become new diagnostic, prognostic or therapeutic biomarkers markers for ccRCC targeted therapy.

## Acknowledgments

The information of this study is obtained by the Oncomine database, GEPIA, UALCAN, cBio-Portal, HPA, TCGA, STRING, GeneMANIA, Metascape and TIMER. We are grateful to them for the source of data used in our study.

## Author Contributions

**Conceptualization:** Tie Chong.

**Data curation:** Mingrui Li, Fangshi Xu.

**Formal analysis:** Mingrui Li, Minxin He, Yibing Guan, Juanhua Tian.

**Investigation:** Fangshi Xu, Yibing Guan, Ziyan Wan, Tie Chong.

**Methodology:** Mingrui Li, Minxin He, Fangshi Xu, Juanhua Tian, Ziyan Wan, Haibin Zhou.

**Project administration:** Yibing Guan.

**Resources:** Mingrui Li, Fangshi Xu.

**Software:** Mingrui Li.

**Supervision:** Mei Gao, Tie Chong.

**Validation:** Juanhua Tian, Haibin Zhou.

**Visualization:** Mingrui Li.

**Writing – original draft:** Mingrui Li.

**Writing – review & editing:** Mingrui Li, Fangshi Xu, Yibing Guan, Mei Gao.

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
