## [Decision Letter · Decision Letter 0]

12 Oct 2021

PONE-D-21-25893Abnormal expression and the significant prognostic value of aquaporins in clear cell renal cell carcinomaPLOS ONE

Dear Dr. Chong,

Thank you for submitting your manuscript to PLOS ONE. After careful consideration, we feel that it has merit but does not fully meet PLOS ONE’s publication criteria as it currently stands. Therefore, we invite you to submit a revised version of the manuscript that addresses the points raised during the review process.

We look forward to receiving your revised manuscript.

Kind regards,

Graça Soveral, PhD

Academic Editor

PLOS ONE

Journal Requirements:

2. We note that Figure 3 in your submission contain copyrighted images. All PLOS content is published under the Creative Commons Attribution License (CC BY 4.0), which means that the manuscript, images, and Supporting Information files will be freely available online, and any third party is permitted to access, download, copy, distribute, and use these materials in any way, even commercially, with proper attribution. For more information, see our copyright guidelines: http://journals.plos.org/plosone/s/licenses-and-copyright.

1. You may seek permission from the original copyright holder of Figure 3 to publish the content specifically under the CC BY 4.0 license. 

2. If you are unable to obtain permission from the original copyright holder to publish these figures under the CC BY 4.0 license or if the copyright holder’s requirements are incompatible with the CC BY 4.0 license, please either i) remove the figure or ii) supply a replacement figure that complies with the CC BY 4.0 license. Please check copyright information on all replacement figures and update the figure caption with source information. If applicable, please specify in the figure caption text when a figure is similar but not identical to the original image and is therefore for illustrative purposes only

Reviewers' comments:

Reviewer's Responses to Questions

**Comments to the Author**

1. Is the manuscript technically sound, and do the data support the conclusions?

Reviewer #1: Partly

Reviewer #2: No

2. Has the statistical analysis been performed appropriately and rigorously? 

Reviewer #1: Yes

Reviewer #2: No

3. Have the authors made all data underlying the findings in their manuscript fully available?

Reviewer #1: Yes

Reviewer #2: No

4. Is the manuscript presented in an intelligible fashion and written in standard English?

Reviewer #1: Yes

Reviewer #2: Yes

5. Review Comments to the Author

Reviewer #1: Li et al. demonstrated the expression of aquaporins (AQPs) and their prognostic value in clear renal cell carcinoma (ccRCC) through bioinformatics analyses. As stated, this study aims to explore the impact of AQPs on the prognosis of ccRCC, and to explore the potential relationship between AQPs and the occurrence and development of ccRCC. Specifically, they explored and examined 1) the expression pattern of AQPs in the ccRCC; 2) protein-protein interaction networking and function enrichment analysis; 3) a comprehensive analysis of the genetic mutant frequency of AQPs in ccRCC; 4) analysis of the main enriched biological functions of AQPs and the correlation with seven main immune cells; and 5) the prognostic value of AQPs.

This study demonstrated a number of interesting findings. However, the biological roles of AQPs in ccRCC were only presented by bioinformatic analyses and the results need to be further confirmed by well-controlled studies. Moreover, the pattern of AQPs expression found in the ccRCC was different from normal kidneys. Furthermore, the authors examined the correlations between the mRNA expression levels of AQPs (not protein levels) and the clinicopathological stages of ccRCC. I have a few major comments, which should be addressed.

Major comments

1. My concern is about the difference in the expression of AQPs between normal kidneys ccRCC. For example, the expression of AQP9,10, 12A, and 12B in normal kidneys is still unknown and was not detected. However, the authors demonstrated their expression and the roles in the ccRCC (for example, lines: 195, 211, 214, 242, and 243). Importantly, the authors demonstrated the genetic mutation of AQPs in ccRCC and stated that enhanced mRNA expression is the most common change in these samples. Then, can we think that AQP9, 10, 12A are abundantly expressed in ccRCC and they have their own functions, which are not expressed in normal kidneys?

2. The authors demonstrated the direct correlation between AQPs expression (AQP1 -12) and the infiltration of immune cells. Is there any background that shows a direct link between AQPs expression levels and the infiltration of immune cells?

3. The authors demonstrated the prognostic value of AQPs in ccRCC, by examining the levels of mRNA expression of AQPs. In general, protein expression levels are affected by post-transcriptional and post-translational modification, and thus protein expression levels are not always the same as their mRNA levels. However, the proteins are mainly responsible for the function. Then how can we interpret the results of mRNA in the clinical setting? Please discuss this issue in the discussion section.

Reviewer #2: Related evidence from the broader field supports the authors' hypothesis that patterns of AQP expression are of interest as diagnostic tools in renal cancers. The overall scope of the review is timely and of interest, but merits further effort to lift the benefit and impact of the contribution to the field. Unfortunately as presented, the work has not yet achieved its potential.

Major concerns:

The Abstract would be more informative for a broad audience if written with minimal use of acronyms. Abbreviations that will be kept in the Abstract must be defined. The Abstract as written is not sufficent in essential details; results ideally would be presented as concrete findings with specific major results, rather than vague statements. Phrases such as "The expression levels of different AQPs members in ccRCC had different trends..." or "This study aims to explore..." have no information content. The most informative approach is to specify the results by AQP class, to state the levels of up- or downregulation ,to report fold changes in risk factors, and to present interpretive summaries of the main findings.

The Methods section briefly summarizes general methods used for data acquisition and analysis. Inclusion of an in-depth Supplementary file for the Methods is recommended highly, fully extending each of the Methods sections (Oncomine, ULACAN, TCGA, GEPIA, cBioPortal, GeneMANIA and other analyses) in order to provide the level of detail that would be necessary for independent replication and verification of the findings.

The Results section should be edited throughout to clarify the lines of argument and to incorporate substantive details with justifications. The overall aim of carrying out a multivariate analysis that draws from multiple databases is laudable, and the focus of this research project is important, but the results as presented have not yet been sufficiently analyzed or coherently integrated to achieve clear and convincing conclusions. As an example, starting just with the first section of the Results (beginning line 161) what precisely were the 20 tumors that were selected, why were only 20 used out of the many samples available in public domain archives, and how were these choices justified? Following on, the presentation of the findings comes across as a random mixture of new observations with previously published findings and thus lacks clarity, as illustrated by statements such as "the expression level of AQP6 was up-regulated or down-regulated in different test results, which was inconsistent. For AQP9/10/12A/12B, no test data could provide suitable results under the established screening conditions." The significance and interpretations of these observations are not clear, and the main points being made are not well articulated.

Unfortunately, the lack of clarity and logic persists throughout the Results section.

For example, in Table 1, the evidence supporting the reported fold changes is not clearly explained. Information in the "Ref" column is not presented in a format that would allow a reader to find the cited work. Which cancers are being compared in this Table is not indicated.

Perhaps overall formatting has been altered inadvertently during document uploading, but the information shown for apparently more than one Table (?) on pages 18-27 seems incomprehensible.

Image quality of the Figures is unacceptably low. Images are blurry and pixelated and should be replaced with higher resolution copies. Some (such as Fig 2) are illegible.

Overall this area of work has promise, but has not yet been sufficiently or carefully developed.

Minor concerns:

References cited need be directly relevant to the text statements, and should acknowledge seminal work when possible. For example, the review by Castle 2005 (7) would not be the best reference for the reported classification of AQP12 subtypes (line 51). Equal care is needed for every citation. All references throughout the MS should be double-checked as the best choices for the matching statements of fact.

Correcting minor errors such as imprecise uses of some words, defining all acronyms at first use, and replacing colloquialisms are small but necessary details to be addressed throughout the text.

6. PLOS authors have the option to publish the peer review history of their article (what does this mean?). If published, this will include your full peer review and any attached files.

Reviewer #1: No

Reviewer #2: No

---

## [Author Response · Author response to Decision Letter 0]

9 Dec 2021

Dear Editors and Reviewers,

Thank you very much for your comments and support. We have revised the manuscript and resubmitted it. Point by point responses to the reviewers’ comments are listed below this letter. Correspondingly, the modifications will also be presented in the revised manuscript. We look forward to hearing from you soon. We also would like to express our sincere thanks to the reviewers and editors for the constructive and positive comments.

Editorial Board Member comments

Comments 1: Please ensure that your manuscript meets PLOS ONE's style requirements, including those for file naming.

Answer 1: Thank you very much for your reminders and suggestions. We apologize for the inconvenience caused to you. We have re-checked the manuscript format to make it meet the submission requirements of plos one.

Comments 2: We note that Figure 3 in your submission contain copyrighted images. All PLOS content is published under the Creative Commons Attribution License (CC BY 4.0), which means that the manuscript, images, and Supporting Information files will be freely available online, and any third party is permitted to access, download, copy, distribute, and use these materials in any way, even commercially, with proper attribution.

Answer 2: This is a very important reminder. We were aware of this copyright issue and immediately contacted the original copyright holder. We have obtained written permission from the copyright owner to publish these figures exclusively under the CC BY 4.0 license. The Content Permission Form has been submitted as an “other” file. The legend of Figure 3 has also been modified as required.

Reviewers' comments

Reviewer #1:

Comments 1: My concern is about the difference in the expression of AQPs between normal kidneys ccRCC. For example, the expression of AQP9,10, 12A, and 12B in normal kidneys is still unknown and was not detected. However, the authors demonstrated their expression and the roles in the ccRCC (for example, lines: 195, 211, 214, 242, and 243). Importantly, the authors demonstrated the genetic mutation of AQPs in ccRCC and stated that enhanced mRNA expression is the most common change in these samples. Then, can we think that AQP9, 10, 12A are abundantly expressed in ccRCC and they have their own functions, which are not expressed in normal kidneys?

Answer 1: Very appreciate for your constructive comments. AQP9 is an important transmembrane protein. After analyses of the UALCAN and HPA, we found that the mRNA and protein expression level of AQP9 in normal kidney tissues and ccRCC tissues were significantly different. AQP9 plays a role in glycerol transport and participates in the regulation of cellular inflammation. It can regulate a series of inflammation-related signal pathways, such as IL6/JAK-STAT3, IL2-STAT5 and TNFα signal pathways, among which the pathway, IL-6/JAK-STAT3, is abnormally activated in many cancers and is usually associated with poor clinical prognosis [1]. In the tumor microenvironment, the IL-6/JAK-STAT3 signaling pathway promotes tumor cell proliferation, invasion and metastasis, while strongly inhibiting tumor immune response [2]. 

Studies of AQP10 in intestinal epithelial cells are relatively comprehensive. In addition to transporting water, AQP10 is also involved in the penetration of ammonia, erythritol and metalloids, including toxic inorganic arsenic and glycerol [3, 4]. Therefore, AQP10 may play a broader unknown role than water transportation. Recent reports on the expression of AQP10 in the stratum corneum and adipocytes of the skin indicated that AQP10 might be widely expressed. However, their western blot showed that there were several AQP10 bands without monomeric bands, in contrast to the clear expression of AQP3 in the skin. This indicated that the expression of AQP10 was the lowest [5]. There was no significant difference in the expression level of AQP10 in normal kidney tissues and ccRCC tissues. The TCGA data showed that compared with other members, the expression level of AQP10 was lower, which is more consistent with its performance in other tissues. 

AQP12A is a subtype of AQP12. Existing studies have shown that AQP12 is mainly expressed in pancreatic tissue, and the inactivation or mutation of AQP12 is closely related to the occurrence of pancreatitis [6]. According to the mRNA expression information in the TCGA database, AQP12A has been detected in some samples of normal kidney tissues and ccRCC tissues, but most of the samples are not detected. This shows that the relationship between AQP12A and kidney or kidney cancer is relatively weaker than that of other members of AQPs. At the same time, the protein expression level of AQP12A is low both in normal tissues and tumor tissues, and there is no significant change in the expression of AQP12A in different stages of ccRCC. The difference indicates that even though there are changes in gene amplification and gene deletion in gene mutation analysis, its role in ccRCC may not be significant.

In general, the difference in the expression level of AQP9 in normal kidney tissues and renal clear cell carcinoma and the related pathways involved in tumors are relatively clear; while AQP10/12A can only detect a certain degree of expression in some normal tissues. However, these members play a certain role in the development of other diseases, and the pathways involved may be related to the occurrence of ccRCC. This requires us to follow up and investigate the mechanism in depth.

Comments 2: The authors demonstrated the direct correlation between AQPs expression (AQP1 -12) and the infiltration of immune cells. Is there any background that shows a direct link between AQPs expression levels and the infiltration of immune cells?

Answer 2: It’s a very professional comment. At present, many studies have confirmed the relationship between AQPs and inflammation. First of all, regarding the expression of immune cells and AQPs, AQPs are expressed in human immune cells of the innate and adaptive immune system [7]. In human blood leukocytes, AQP1 and AQP9 were detected and upregulated after intravenous or in vitro lipopolysaccharide stimulation [8]. AQP9 is also increased in activated polymorphonuclear leukocytes in patients with systemic inflammatory response syndrome and infective endocarditis [9, 10]. Basic studies have found that AQP1, AQP3, and AQP5 are detected in tumor-infiltrating lymphocytes around lung cancer, and AQP3 and AQP5 are detected in dendritic cells. Their expression is related to the activation and proliferation of these immune cells [11].

The second is the cytokine and growth factor signaling pathways that AQPs participate in inflammation. In vitro experiments have shown that inhibition or silencing of AQP3 will partially block LPS initiation and reduce the production of interleukin IL-6, IL-1β and TNF-α, indicating that there is a connection between AQP3 function and TLR4 in the initiation of macrophages [12, 13]. Another study reports that AQP1 expression increases after LPS administration, while AQP5 mRNA expression decreases [14]. In addition, patients with systemic inflammatory response syndrome have increased AQP9 expression in neutrophils compared to healthy controls [9].

Finally, it is about the regulation of immune cells by AQPs. Several studies have shown that AQPs regulate the migration of different immune cells. For example, AQP3 is essential for T cell function and macrophage migration, while AQP5 and AQP9 regulate neutrophil migration and affect the survival of sepsis [15, 16].

In general, AQPs do play an important role in inflammation by regulating immune cell migration and participating in inflammation-related cytokine pathways. Studies have found the expression of AQPs in infiltrating immune cells in the tumor microenvironment of lung cancer. However, there is still a lack of relevant research on kidney cancer.

Comments 3: The authors demonstrated the prognostic value of AQPs in ccRCC, by examining the levels of mRNA expression of AQPs. In general, protein expression levels are affected by post-transcriptional and post-translational modification, and thus protein expression levels are not always the same as their mRNA levels. However, the proteins are mainly responsible for the function. Then how can we interpret the results of mRNA in the clinical setting? Please discuss this issue in the discussion section.

Answer 3: Thank you for your reminder, this is a valuable suggestion. What we consider is that if there is no change in mRNA expression, the possibility of protein change is very low, because the basis of the translation process is dependent on the transcription process [17, 18]. Therefore, mRNA expression changes can be used as a method to find DEGs preliminary screening, but as you said, further verification of protein expression is needed. Following your suggestion, we discussed this issue in the discussion section and marked it with the "Track Changes" function.

Reviewer #2:

Comments 1: The Abstract would be more informative for a broad audience if written with minimal use of acronyms. Abbreviations that will be kept in the Abstract must be defined. The Abstract as written is not sufficent in essential details; results ideally would be presented as concrete findings with specific major results, rather than vague statements. Phrases such as "The expression levels of different AQPs members in ccRCC had different trends..." or "This study aims to explore..." have no information content. The most informative approach is to specify the results by AQP class, to state the levels of up- or downregulation ,to report fold changes in risk factors, and to present interpretive summaries of the main findings.

Answer 1: It’s a very meaningful comment. We have provided a complete definition of the abbreviations in the abstract to ensure that readers can correctly understand the meaning of the abbreviations. Through your guidance, we have understood that the abstract is improperly written and there are many repetitions. We have revised the places you specifically pointed out, and also revised other parts of the abstract. The abstract part has been marked with the "Track Changes" function.

Comments 2: The Methods section briefly summarizes general methods used for data acquisition and analysis. Inclusion of an in-depth Supplementary file for the Methods is recommended highly, fully extending each of the Methods sections (Oncomine, ULACAN, TCGA, GEPIA, cBioPortal, GeneMANIA and other analyses) in order to provide the level of detail that would be necessary for independent replication and verification of the findings.

Answer 2: It’s a very constructive comment. We have supplemented the steps involved in the method section. The modified parts have been marked by the "Track Changes" function.

Comments 3: The Results section should be edited throughout to clarify the lines of argument and to incorporate substantive details with justifications. The overall aim of carrying out a multivariate analysis that draws from multiple databases is laudable, and the focus of this research project is important, but the results as presented have not yet been sufficiently analyzed or coherently integrated to achieve clear and convincing conclusions. As an example, starting just with the first section of the Results (beginning line 161) what precisely were the 20 tumors that were selected, why were only 20 used out of the many samples available in public domain archives, and how were these choices justified? Following on, the presentation of the findings comes across as a random mixture of new observations with previously published findings and thus lacks clarity, as illustrated by statements such as "the expression level of AQP6 was up-regulated or down-regulated in different test results, which was inconsistent. For AQP9/10/12A/12B, no test data could provide suitable results under the established screening conditions." The significance and interpretations of these observations are not clear, and the main points being made are not well articulated.

Answer 3: Very appreciate for your comments. In the first part of the results, we presented the results of the difference in mRNA expression of AQPs in pan-cancer in the Oncomine database. As a product adapted to the big data analysis environment, Oncomine brings together 715 data sets and 86733 samples of different cancers in the human body. Cancer categories are bladder cancer, brain and CNS cancer, breast cancer, cervical cancer, colorectal cancer, esophageal cancer, gastric cancer, head and neck cancer, kidney cancer, leukemia, liver cancer, lung cancer, lymphoma, melanoma, myeloma, ovarian cancer, pancreatic cancer, prostate cancer, sarcoma and other cancer. The Oncomine platform uses these as categories to visualize the mRNA expression changes of different human genes in pan-cancers (Fig 1), so we called it ‘in 20 different tumors’ in the article. At the same time, we wanted to show the visualization results of the changes in the expression of AQPs in different datasets of kidney cancer through Figure 1. The color of the square represented the change trend (red-up regulation, blue-down regulation), and the number represented datasets meeting the screening criteria (gene rank, 10%; fold change, 2; and P-value, 0.01). At the same time, we found that these datasets were not all related to ccRCC, so we screened these datasets one by one and made detailed statistics and data source references for the related datasets of ccRCC (Table 1). We described in the article that " the expression level of AQP6 was up-regulated or down-regulated in different test results, which was inconsistent. " was also the reason for the datasets. Among AQPs, only the expression differences of AQP6 had opposite trends in different datasets. After reading the dataset information related to AQP6, we found that the datasets showing the up-regulation of AQP6 expression in kidney cancer tissues were not the datasets related to ccRCC, and the datasets that obtained the expression down-regulation result were the ccRCC datasets. This result is consistent with the result of the subsequent UALCAN database analysis (Fig 2). And AQP9/10/12A/12B did not have any datasets that could meet the established screening conditions (gene rank, 10%; fold change, 2; and P-value, 0.01). If the screening conditions were lowered, any datasets that meet the conditions could be obtained, but this would result in a lack of diversity in the information obtained from other genes.

Comments 4: Unfortunately, the lack of clarity and logic persists throughout the Results section. For example, in Table 1, the evidence supporting the reported fold changes is not clearly explained. Information in the "Ref" column is not presented in a format that would allow a reader to find the cited work. Which cancers are being compared in this Table is not indicated.

Answer 4: Thank you for your reminding. Table 1 is a detailed summary of the datasets related to ccRCC in the results of the Oncomine database (see the header of Table 1). The fold changes of the difference between AQPs in ccRCC and normal kidney tissues are the real data obtained from the Oncomine database. The relevant documents involved in the ‘Ref’ column were listed in the text ‘Next, we analyzed the specific data (Table 1)[23],[24],[25],[26],[27], [28].’

Comments 5: Perhaps overall formatting has been altered inadvertently during document uploading, but the information shown for apparently more than one Table (?) on pages 18-27 seems incomprehensible.

Answer 5: Many thanks for your comments. A total of 6 tables are shown on pages 18-27. Among them, Table2 is the baseline data table of 526 cases of ccRCC samples, and Table3 is the result of univariate Cox regression analysis of AQPs, indicating that AQP0/1/4/9 are prognostic related factors. Table4-7 are the results of multivariate Cox regression analysis of AQP0/1/4/9, respectively, indicating that AQP1/9 are independent prognostic factors. We made a mistake by treating table titles as table legends and then writing table titles below tables, which caused you to be unable to understand the content of the table correctly. Now we have revised all the tables in the manuscript. The modified parts have been marked by the "Track Changes" function.

Comments 6: Image quality of the Figures is unacceptably low. Images are blurry and pixelated and should be replaced with higher resolution copies. Some (such as Fig 2) are illegible. Overall this area of work has promise, but has not yet been sufficiently or carefully developed.

Answer 6: Thank you for your reminding. We have re-made the picture, and set the resolution, file size, font, compression method and other aspects based on the picture guide of plos one.

Comments 7: References cited need be directly relevant to the text statements, and should acknowledge seminal work when possible. For example, the review by Castle 2005 (7) would not be the best reference for the reported classification of AQP12 subtypes (line 51). Equal care is needed for every citation. All references throughout the MS should be double-checked as the best choices for the matching statements of fact.

Answer 7: We have changed the wrong citation, replaced with "The role of mammalian superaquaporins inside the cell: An update" (PMID:33811846). We also checked the rationality of other documents again.

Comments 8: Correcting minor errors such as imprecise uses of some words, defining all acronyms at first use, and replacing colloquialisms are small but necessary details to be addressed throughout the text.

Answer 8: Many thanks for your recommendation. For imprecise uses of some words, defining all acronyms at first use, and replacing colloquialisms and other questions, we have reviewed the full text and made several revisions. The modified parts have been marked by the "Track Changes" function.

References:

1. Xu WH, Shi SN, Xu Y, Wang J, Wang HK, Cao DL, et al. Prognostic implications of Aquaporin 9 expression in clear cell renal cell carcinoma. J Transl Med. 2019;17(1):363.

2. Johnson DE, O'Keefe RA, Grandis JR. Targeting the IL-6/JAK/STAT3 signalling axis in cancer. Nat Rev Clin Oncol. 2018;15(4):234-48.

3. Laforenza U, Scaffino MF, Gastaldi G. Aquaporin-10 represents an alternative pathway for glycerol efflux from human adipocytes. PLoS One. 2013;8(1):e54474.

4. Litman T, Sogaard R, Zeuthen T. Ammonia and urea permeability of mammalian aquaporins. Handb Exp Pharmacol. 2009(190):327-58.

5. Jungersted JM, Bomholt J, Bajraktari N, Hansen JS, Klaerke DA, Pedersen PA, et al. In vivo studies of aquaporins 3 and 10 in human stratum corneum. Arch Dermatol Res. 2013;305(8):699-704.

6. Ishibashi K, Tanaka Y, Morishita Y. The role of mammalian superaquaporins inside the cell: An update. Biochim Biophys Acta Biomembr. 2021;1863(7):183617.

7. Talwar S, Munson PJ, Barb J, Fiuza C, Cintron AP, Logun C, et al. Gene expression profiles of peripheral blood leukocytes after endotoxin challenge in humans. Physiol Genomics. 2006;25(2):203-15.

8. Vassiliou AG, Maniatis NA, Orfanos SE, Mastora Z, Jahaj E, Paparountas T, et al. Induced expression and functional effects of aquaporin-1 in human leukocytes in sepsis. Crit Care. 2013;17(5):R199.

9. Matsushima A, Ogura H, Koh T, Shimazu T, Sugimoto H. Enhanced expression of aquaporin 9 in activated polymorphonuclear leukocytes in patients with systemic inflammatory response syndrome. Shock. 2014;42(4):322-6.

10. Thuny F, Textoris J, Amara AB, Filali AE, Capo C, Habib G, et al. The gene expression analysis of blood reveals S100A11 and AQP9 as potential biomarkers of infective endocarditis. PLoS One. 2012;7(2):e31490.

11. Moon C, Rousseau R, Soria JC, Hoque MO, Lee J, Jang SJ, et al. Aquaporin expression in human lymphocytes and dendritic cells. Am J Hematol. 2004;75(3):128-33.

12. da Silva IV, Cardoso C, Martinez-Banaclocha H, Casini A, Pelegrin P, Soveral G. Aquaporin-3 is involved in NLRP3-inflammasome activation contributing to the setting of inflammatory response. Cell Mol Life Sci. 2021;78(6):3073-85.

13. Rabolli V, Wallemme L, Lo Re S, Uwambayinema F, Palmai-Pallag M, Thomassen L, et al. Critical role of aquaporins in interleukin 1beta (IL-1beta)-induced inflammation. J Biol Chem. 2014;289(20):13937-47.

14. Rump K, Brendt P, Frey UH, Schafer ST, Siffert W, Peters J, et al. Aquaporin 1 and 5 expression evoked by the beta2 adrenoreceptor agonist terbutaline and lipopolysaccharide in mice and in the human monocytic cell line THP-1 is differentially regulated. Shock. 2013;40(5):430-6.

15. Rump K, Adamzik M. Function of aquaporins in sepsis: a systematic review. Cell Biosci. 2018;8:10.

16. Hara-Chikuma M, Chikuma S, Sugiyama Y, Kabashima K, Verkman AS, Inoue S, et al. Chemokine-dependent T cell migration requires aquaporin-3-mediated hydrogen peroxide uptake. J Exp Med. 2012;209(10):1743-52.

17. Mercadante AA, Dimri M, Mohiuddin SS. Biochemistry, Replication and Transcription. StatPearls. Treasure Island (FL)2021.

18. Buccitelli C, Selbach M. mRNAs, proteins and the emerging principles of gene expression control. Nat Rev Genet. 2020;21(10):630-44.

---

## [Decision Letter · Decision Letter 1]

6 Jan 2022

PONE-D-21-25893R1Abnormal expression and the significant prognostic value of aquaporins in clear cell renal cell carcinomaPLOS ONE

Dear Dr. Chong,

Thank you for submitting your manuscript to PLOS ONE. After careful consideration, we feel that it has merit but does not fully meet PLOS ONE’s publication criteria as it currently stands. Therefore, we invite you to submit a revised version of the manuscript that addresses the points raised during the review process.

Specifically, reviewer 2 still suggests improvements and the questions raised by reviewer 1 were not fully answered or elucidated. I recommend you address all the points in detail and insert the discussed topics into the manuscript to clarify the interpretation and significance of your data. You should indicate all the modifications in the rebuttal letter.

We look forward to receiving your revised manuscript.

Kind regards,

Graça Soveral, PhD

Academic Editor

PLOS ONE

Reviewers' comments:

Reviewer's Responses to Questions

**Comments to the Author**

1. If the authors have adequately addressed your comments raised in a previous round of review and you feel that this manuscript is now acceptable for publication, you may indicate that here to bypass the “Comments to the Author” section, enter your conflict of interest statement in the “Confidential to Editor” section, and submit your "Accept" recommendation.

Reviewer #1: (No Response)

Reviewer #2: All comments have been addressed

2. Is the manuscript technically sound, and do the data support the conclusions?

Reviewer #1: No

Reviewer #2: Yes

3. Has the statistical analysis been performed appropriately and rigorously? 

Reviewer #1: Yes

Reviewer #2: Yes

4. Have the authors made all data underlying the findings in their manuscript fully available?

Reviewer #1: No

Reviewer #2: Yes

5. Is the manuscript presented in an intelligible fashion and written in standard English?

Reviewer #1: Yes

Reviewer #2: Yes

6. Review Comments to the Author

Reviewer #1: Although research project in this study is interesting, I feel that the results have not yet been sufficiently analyzed, or integrated to achieve clear and convincing conclusions. Moreover, any cellular, molecular, and biological experiments were not done to support the author's tentative conclusions. As an example, I concerned about the differences in the expression of AQPs between normal human kidneys and ccRCC. Specifically, I asked about the expression of AQP9, 10, 12A in the kidneys, and the authors should test and compare the expression levels of these AQPs by simple approaches, e.g, PCR or protein experiments. In addition, AQP0, 5, 8 should also be tested since these AQPs are not present in normal kidneys. Thus, I think that the significance and interpretations out of the authors' observations are not clear, and the main points being made are not conclusive.

Reviewer #2: The MS has been greatly improved by the authors' earnest and detailed responses to the reviewers' comments, and successfully presents a comprehensive and informative analysis of the correlations of patterns of AQP expression with many RCC properties.

To achieve best impact, several small details are recommended to be addressed, as listed below:

Table 1. The "Ref" column is still not clearly utilized. List reference numbers for the published papers instead of "name_Renal".

In the Fig 1 legend, the term "Differences in transcriptional expression" is vague. Precisely define the numbers shown in the cells, and the meanings of the fill colors.

Fig 3. It would be wise to double-check whether the antibodies used in the Human Atlas database are capable of distinguishing between isoforms AQP12A and 12B. If available antibodies recognize a common epitope, then showing this analysis as separate sets is somewhat misleading. In that case, perhaps the four images should be combined under one subheading '12 A&B' or similar.

Fig 4. The legend needs to include a brief description of the type of plot being used to display the data, and definitions of terms and abbreviations shown in the panel keys.

Fig 5. Similar to comments above for Fig 4, more details are needed in the legend to define the data shown in each panel.

Text in the Results for Fig 6 would benefit from adding an interpretive summary statement at the end of this section that captures the main outcome or major finding that has been shown as a result of the multiple analyses.

In the results text for Fig 7, the extensive numerical details could be moved to a table, and the text focused more clearly on explaining the data shown, with an interpretive summary.

Results for Fig 8 are said to be aimed at "analyzing AQPs as a whole", but the justification for this is opaque. The Figure seems unnecessary given the extended details provided in the subsequent Fig 9.

Figs 9 and 10. Using the same headings "overall survival" or "disease free survival" for all panels is not informative. The legend could state what type of survival is plotted, and the panel headings then more helpfully could list the AQP class being studied.

The data appear to show that AQP1 levels are the most important predictor of survival, as compared to relatively smaller effects of other AQPs, but this idea (if correct) is not clearly evaluated as a possible principal outcome of the study in the Discussion or Conclusions.

Minor grammar errors throughout could use polishing by a language-fluent editor to showcase the work as effectively as possible, but it is understandable as written.

7. PLOS authors have the option to publish the peer review history of their article (what does this mean?). If published, this will include your full peer review and any attached files.

Reviewer #1: No

Reviewer #2: No

---

## [Author Response · Author response to Decision Letter 1]

8 Feb 2022

(Some figures and tables are included in our response letter, but cannot be displayed here, please see the document "Response to reviewers" for details.)

Dear Editors and Reviewers,

Thanks very much for taking your time to review our manuscript "Abnormal expression and the significant prognostic value of aquaporins in clear cell renal cell carcinoma" (Manuscript Number: PONE-D-21-25893R1). We really appreciate all your comments and suggestions! Your suggestions have enabled us to improve our work. We have resubmitted the file of the revised manuscript with highlighted changes. We have also responded to the comments raised by the reviewers in a point-to-point response manner in this letter. We would like also to thank you for allowing us to resubmit a revised copy of the manuscript.

Editor comment

Comment: Specifically, reviewer 2 still suggests improvements and the questions raised by reviewer 1 were not fully answered or elucidated. I recommend you address all the points in detail and insert the discussed topics into the manuscript to clarify the interpretation and significance of your data. You should indicate all the modifications in the rebuttal letter.

Answer: Many thanks for the editor’ s constructive and professional comments. We have carefully reviewed the relevant literature according to two reviewers’ comments and revised the manuscript accordingly. Particular attention was fixed on interpretation and significance of our findings. Additionally, we added the limitations of our study in the revised manuscript. For details, please refer to the replies to the two reviewers. We sincerely hope that the revised manuscript would meet PLOS ONE’s publication criteria.

Reviewers' comments

Reviewer #1:

Comment: Although research project in this study is interesting, I feel that the results have not yet been sufficiently analyzed, or integrated to achieve clear and convincing conclusions. Moreover, any cellular, molecular, and biological experiments were not done to support the author's tentative conclusions. As an example, I concerned about the differences in the expression of AQPs between normal human kidneys and ccRCC. Specifically, I asked about the expression of AQP9, 10, 12A in the kidneys, and the authors should test and compare the expression levels of these AQPs by simple approaches, e.g, PCR or protein experiments. In addition, AQP0, 5, 8 should also be tested since these AQPs are not present in normal kidneys. Thus, I think that the significance and interpretations out of the authors' observations are not clear, and the main points being made are not conclusive.

Answer: Very appreciate for your constructive comments. We are very sorry that we did not adequately answer your questions in our last response letter. We have re-reviewed the previous comments and combine them with the comments this time, and have learned that your focus is on the expression level of AQP0/5/8/9/10/12A in the normal kidney tissue. We are very sorry that due to the recent repeated or even exacerbated COVID-19 epidemic in Xi'an, China, our laboratory cannot be opened normally for the time being, so we are afraid we cannot verify the expression level of AQPs in the normal kidney tissue by molecular biology. Therefore, we actively try to solve this problem using alternatives. Genotype-tissue expression (GTEx) project is one of the largest resources, containing about 11,600 RNA-sequenced tissue profiles from over 40 human tissues. We downloaded gene expression and annotation files from the GTEx database and visualized the expression of AQPs in the normal human tissues by R language. As is shown in S1 Fig (see in the next page), AQP0/5/8/9/10/12A are all expressed in the normal kidney tissue to varying degrees although some of them have low expression levels. At the same time, combined with the result in the UALCAN database, we believe that AQPs are expressed in the normal kidney tissue at the transcriptional level (Fig 2). We hope that, to a certain extent, these results could eliminate your confusion about the expression of AQPs in the normal kidney tissue as well as make our results more convincing. We are truly sorry for our inability to conduct the experiment in the short term. We are also aware of this deficiency so we have demonstrated it as the "limitation" in the revised manuscript. And we intend to further explore the expression and molecular mechanism of the members of AQPs in the normal kidney tissue and ccRCC by molecular biological methods when conditions permit in the future.

S1 Fig. Expressions of AQP0/5/8/9/10/12A in the normal kidney tissue (GTEx). The GTEx database provides information on 104 normal kidney tissue samples, and the word "kidney" has been bolded and highlighted in red in order to emphasize the expressions of AQP0/5/8/9/10/12A in the normal kidney tissue. The result of GTEx supplements the expression levels of AQPs in the normal kidney to some extent.

Fig 2. Expressions of AQPs in ccRCC and adjacent normal kidney tissues (Ualcan). 

Supplements in the text (limitations):

Our study provides a comprehensive analysis of AQPs in ccRCC using multiple online databases based on the most popular bioinformatics theories. These available methods are characterized by large sample size, low cost, and the ability to perform large-scale genomics studies and functional analyses. However, our study still has some limitations. First, online databases have limitations. Different databases may yield different results due to differences in the sources of the collected samples. Furthermore, this study only presents the results of bioinformatics analysis based on different online databases. Therefore, further molecular biology experiments, including quantitative real time polymerase chain reaction (qRT-PCR), Western blot, and immunohistochemistry, are required to validate the results of this study.

Reviewer #2:

Comments 1: Table 1. The "Ref" column is still not clearly utilized. List reference numbers for the published papers instead of "name_Renal".

Answer 1: It’s a very meaningful comment. Thanks to your guidance, we have made corresponding modifications in Table 1. The revised table is shown below.

Changes in the table:

Table 1. Significant changes of AQPs expression in transcription level between ccRCC and normal kidney tissues (Oncomine).

 Types of ccRCC VS. Kidney Fold Change p-value t-test Ref

AQP1 

 Non-Hereditary Clear Cell Renal Cell Carcinoma -4.503 1.70E-05 -4.751 Beroukhim Renal [23]

AQP2 

 Clear Cell Renal Cell Carcinoma -5.590 2.93E-11 -13.694 Gumz Renal [24]

 Clear Cell Renal Cell Carcinoma -2.567 5.55E-6 -6.779 Lenburg Renal [25]

 Clear Cell Renal Cell Carcinoma -13.183 0.002 -5.550 Yusenko Renal [26]

 Clear Cell Renal Cell Carcinoma -4.592 1.46E-10 -10.233 Jones Renal [27]

 Non-Hereditary Clear Cell Renal Cell Carcinoma -11.212 1.87E-08 -12.468 Beroukhim Renal [23]

 Hereditary Clear Cell Renal Cell Carcinoma -12.336 4.06E-08 -13.460 Beroukhim Renal [23]

AQP3 

 Clear Cell Renal Cell Carcinoma -3.889 4.56E-6 -7.538 Higgins Renal [28]

 Non-Hereditary Clear Cell Renal Cell Carcinoma -3.699 8.93E-10 -8.449 Beroukhim Renal [23]

 Hereditary Clear Cell Renal Cell Carcinoma -2.391 9.75E-07 -5.931 Beroukhim Renal [23]

AQP6 

 Clear Cell Renal Cell Carcinoma -4.140 1.18E-11 -14.586 Gumz Renal [24]

 Clear Cell Renal Cell Carcinoma -8.330 1.01E-4 -7.220 Yusenko Renal [26]

AQP7 

 Clear Cell Renal Cell Carcinoma -9.610 4.45E-07 -9.447 Gumz Renal [24]

 Clear Cell Renal Cell Carcinoma -2.102 3.26E-04 -4.863 Lenburg Renal [25]

AQP8 

 Clear Cell Renal Cell Carcinoma -3.097 2.52E-30 -29.476 Jones Renal [27]

Comments 2: In the Fig 1 legend, the term "Differences in transcriptional expression" is vague. Precisely define the numbers shown in the cells, and the meanings of the fill colors.

Answer 2: It’s a very constructive comment. We have supplemented the specific meaning of the numbers and colors in Fig 1. The revised Fig 1 caption is described as follow.

Changes in the figure caption:

Fig 1. Transcriptional expression of AQPs in 20 different types of cancer diseases (Oncomine). The numbers in the small boxes represent the number of datasets with statistically significant mRNA differences of the target genes, in which red represents over-expression and blue represents down-regulated expression. Differences in transcriptional expression between ccRCC and normal kidney tissues were compared by students’ t-test. The cut off of p-value and fold change were as following: p-value: 0.01, fold change: 2, gene rank: 10%, data type: mRNA.

Comments 3: Fig 3. It would be wise to double-check whether the antibodies used in the Human Atlas database are capable of distinguishing between isoforms AQP12A and 12B. If available antibodies recognize a common epitope, then showing this analysis as separate sets is somewhat misleading. In that case, perhaps the four images should be combined under one subheading '12 A&B' or similar.

Answer 3: Very appreciate for your comments. Your sensitive perception is correct. We re-reviewed the description in the manuscript and repeated the analysis in the HPA database. The antibodies that detected AQP12A and AQP12B are indeed identical. Therefore, we have modified Fig 3 and the figure caption as you suggested. The revised figure and figure caption are as follows.

Changes in the figure and figure caption:

Fig 3. Representative immunohistochemistry images of AQPs in ccRCC and normal kidney tissues (Human Protein Atlas). (A, C, D, G - I) AQP0/2/3/6/8/9 proteins were not expressed in ccRCC tissues, whereas their low, medium and high expressions were observed in normal kidney tissues. (B, E) Low protein expressions of AQP1/4 were found in ccRCC tissues, while their medium protein expressions were observed in normal kidney tissues. (F, J - K) AQP5/10/12A&B proteins were expressed neither in ccRCC nor normal kidney tissues. Republished from https://www.proteinatlas.org/ under a CC BY license, with permission from inger åhlén, original copyright 2021.

Comments 4: Fig 4. The legend needs to include a brief description of the type of plot being used to display the data, and definitions of terms and abbreviations shown in the panel keys.

Answer 4: Thanks for your reminding. We have modified the caption of Fig 4 based on your suggestion. The modification is as follow.

Changes in the figure caption:

Fig 4. Relationship between mRNA expression of AQPs and individual cancer stages of ccRCC patients (GEPIA). The figure was visualized using violin plots. mRNA expression levels of AQP1/2/4/6/7/9 were remarkably correlated with patients' individual cancer stages, while the expression across stages did not significantly differ for AQP0/3/5/8/10/11/12A/12B. The method for differential expression gene analysis was one-way ANOVA, using the pathological stage as the variable for calculating differential expression. The F-value was the test statistic from the F-test. The Pr (>F) was the p-value of the F-test and the p-value < 0.05 was significant.

Comments 5: Fig 5. Similar to comments above for Fig 4, more details are needed in the legend to define the data shown in each panel.

Answer 5: Many thanks for your comments. We're so sorry about this. We have modified the caption of Fig 5 to make the information clearer. The modification is as follow.

Changes in the figure caption:

Fig 5. Genetic mutations in AQPs and their association with the OS and DFS of ccRCC patients (cBioPortal). (A) The bar plot showed the overall alterations of AQPs in the ccRCC samples. The vertical axis represented the percentage of alterations, and the horizontal axis represented the selected tumor sample dataset (ccRCC). Among them, the green indicated mutation, the red was amplification, the blue was deep deletion, the pink was high mRNA expression, and the gray was multiple alterations. A high mutation rate (33.18%) could be observed in ccRCC patients. (B) Based on a query of AQPs, cBioPortal summarized genomic alterations in all queried genes across the ccRCC sample set. Each row represented a gene, and each column represented a tumor sample. Brown squares indicated inframe mutations, green squares were missense mutations, gray squares were truncating mutations, red bars were gene amplifications, blue bars were deep deletions, and gray bars with red borders were high mRNA expressions. Mutation rates of each gene were 0%, 5%, 2.5%, 1.8%, 3%, 2.9%, 0.7%, 6%, 0.9%, 4%, 1.1%, 2.7%, 6% and 7%, respectively. (C) The figure showed the correlation between AQPs in ccRCC. For each queried gene, Pearson’s correlation coefficients were calculated and displayed in the corresponding boxes. (D - E) The figure indicated the survival difference between altered and unaltered groups. Genetic alterations of AQPs were not associated with the OS or DFS of ccRCC patients.

Comments 6: Text in the Results for Fig 6 would benefit from adding an interpretive summary statement at the end of this section that captures the main outcome or major finding that has been shown as a result of the multiple analyses.

Answer 6: Thanks for your suggestions and guidance. We are aware of the lack of summative descriptions and explanations for the text of Fig 6. Therefore, we have made corresponding supplements as follows.

Supplements in the figure text:

Overall, AQPs played a significant role in the assembly, trafficking and regulation of gap junctions between cells. Through this special intercellular connection, AQPs regulate the intercellular transport of water, glycerol, carbon dioxide and other substances, thereby maintaining the normal physiological state of cells.

Comments 7: In the results text for Fig 7, the extensive numerical details could be moved to a table, and the text focused more clearly on explaining the data shown, with an interpretive summary.

Answer 7: Thanks a lot for your valuable suggestions. We have realized that the text for Fig 7 is too cumbersome and lacks of focus. Therefore, based on your suggestions, we have made a table to fill in numerical details, and explained the data shown in detail in the text. We are so sorry that our descriptions of the figures were often not rigorous enough, so we also supplemented the caption of the Fig 7 and corresponding text content with more descriptive and conclusive interpretations. Since the new table is located between the previous Table 1 and Table 2, we have rearranged the order of the tables in the revised manuscript. The modifications are as follows.

Supplements in the table:

Table 2. The correlations of AQPs expression with immune infiltration level in ccRCC.

genes B cells CD8+T cells CD4+T cells macrophages neutrophils dendritic cells

AQP0 0.057 -0.04 0.038 0.025 0.087 0.02

AQP1 0.027 0.215 0.065 0.062 0.063 0.051

AQP2 -0.025 -0.049 0.002 -0.053 -0.148 -0.03

AQP3 0.187 0.137 0.045 0.164 0.121 0.097

AQP4 -0.029 0.116 0.101 0.197 0.176 0.057

AQP5 -0.147 -0.052 0.124 -0.048 -0.083 -0.078

AQP6 -0.111 -0.146 -0.037 -0.111 -0.128 -0.126

AQP7 -0.072 -0.029 -0.067 -0.072 -0.112 -0.101

AQP8 -0.158 -0.062 0.101 -0.03 0.038 -0.069

AQP9 0.113 -0.016 0.032 0.269 0.29 0.166

AQP10 -0.053 0.169 0.166 0.11 0.112 0.062

AQP11 0.323 0.008 0.009 0.171 0.109 0.16

AQP12A 0.019 0.033 0.009 0.05 0.007 0.034

AQP12B 0.03 0.044 0.073 0.097 0.057 0.082

The table showed the associations between individual AQP genes and different immune infiltration by listing the corrected spearman correlation coefficient. The bold values indicated statistical significance (p-value < 0.05).

Changes in the figure caption:

Fig 7. Correlations of B cells, CD8+T cells, CD4+T cells, macrophages, neutrophils, dendritic cells and AQPs in ccRCC (TIMER). Correlations of mRNA expression levels between cell markers of B cells, CD8+T cells, CD4+T cells, macrophages, neutrophils, dendritic cells and AQPs in ccRCC patients were shown. The ‘partial cor’ indicated the purity-corrected partial Spearman’s rho value, and the ‘p’ referred to ‘p-value’ (p-value < 0.05 was significant).

Changes in the text:

Considering that inflammation and infiltrating immune cells would affect the prognosis of ccRCC, we used the TIMER database to comprehensively study the correlation between differentially expressed AQPs and the infiltration of immune cells (Fig 7 and Table 2). It could be found that not every gene of AQPs was closely related to the immune cell infiltration. Individuals with significant immune infiltration correlations were AQP3/4/6/9/10/11. There were also differences in the expression levels of mRNA and protein of these genes in ccRCC, which might mean that there was a certain relationship among the abnormal expression levels of AQPs and the resulting inflammatory microenvironment as well as tumor progression. At the same time, we could observe that AQP3/4/9/10/11 were mainly positively correlated with each immune cell, while AQP6 was negatively correlated with the infiltration degree of immune cells, which suggested that the effects of different AQPs in ccRCC might not be completely consistent. Next, it could be noticed that each gene of AQPs was primarily associated with the infiltration of B cells, macrophages and neutrophils, which indicated that these three immune cells might played a major role in the progression of ccRCC.

Comments 8: Results for Fig 8 are said to be aimed at "analyzing AQPs as a whole", but the justification for this is opaque. The Figure seems unnecessary given the extended details provided in the subsequent Fig 9.

Answer 8: Many thanks for your recommendation. Your suggestion is correct. The prognostic correlation of overall AQPs shown in Fig 8 is indeed less meaningful than the prognostic status of individual genes illustrated in Fig 9 . Therefore, we decided to delete the Fig 8 and modify the serial numbers of Fig 9 and Fig10 (become the new Fig 8 and Fig 9).

Comments 9: Figs 9 and 10. Using the same headings "overall survival" or "disease free survival" for all panels is not informative. The legend could state what type of survival is plotted, and the panel headings then more helpfully could list the AQP class being studied.

Answer 9: Thank you for your professional direction. We have redrawn the Fig 9 and Fig 10 (since we have removed the previous Fig 8, their new serial numbers are Fig 8 and Fig 9). The modifications are as follows.

Changes in the figures:

 Fig 8. Prognostic significance of AQPs for the OS of ccRCC patients (GEPIA).

 Fig 9. Prognostic significance of AQPs for the DFS of ccRCC patients (GEPIA).

Comments 10: The data appear to show that AQP1 levels are the most important predictor of survival, as compared to relatively smaller effects of other AQPs, but this idea (if correct) is not clearly evaluated as a possible principal outcome of the study in the Discussion or Conclusions.

Answer 10: Thanks for your suggestions. Your idea is correct. Through our multiple analyses and comprehensive consideration, AQP1 does play a significant role in ccRCC and should be further elaborated. At the same time, we also would like to supplement the explanation of AQP9 in the manuscript, because AQP9 showed equally significant effects, whether in expression level, gene mutation status, immune cell infiltration, or prognostic relevance. Therefore, we reviewed the relevant literature and further elaborated on AQP1/9 in the Discussion and also summarized it in the Conclusions. The changes are as follows.

Supplements in the text (the Discussion):

Finally, it is worthy to pay special attention to the independent prognostic factors AQP1 and AQP9 that are finally screened. Through multi-dimensional analysis, it is not difficult to find that there are significant changes in the mRNA expression levels of AQP1 and AQP9 in ccRCC, and both will change significantly with the progression of ccRCC stages. It is worth noting, however, that AQP1 and AQP9 have different trends in these respects. The expression level of AQP1 in ccRCC shows a gradually decreasing trend, while that of AQP9 is the opposite. In terms of mRNA mutation and immune cell infiltration, the expressions of AQP1 and AQP9 are relatively consistent. The mRNA expression levels of AQP1 and AQP9 are positively correlated with the infiltration of most immune cells. Clinical prognosis correlation analysis indicated that the group with a high expression level of AQP1 tends to show better OS and DFS, while the group with a high expression level of AQP9 means poorer OS and DFS. It is well known that the occurrence of most kidney cancers is closely related to the abnormal function of the von Hippel-Lindau (VHL) gene. The abnormality of the VHL gene can lead to the reduction and accumulation of hypoxia-inducible factor (HIF), thereby initiating the transcriptional activation of hypoxia-responsive genes, and ultimately promoting the occurrence and development of cancer [76]. In the pathway AQP1 is involved in, the up-regulation of AQP1 can promote the stabilization of HIF, which in turn makes the high expression group show a better prognosis [33]. Jing JB et al. explored the expression of AQP9 which was consistent with ours, and they also analyzed the relationship between AQP9 and tumor environment. The results showed that in ccRCC, AQP9 promoted tumor-associated macrophage polarization, inhibited the recruitment of natural killer (NK) cells and CD8+ T cells by inhibiting P53 and activating the Janus kinase/signal transducer and activator of transcription (JAK/STAT) pathway, and ultimately stimulated a reorientation of the tumor microenvironment in ccRCC toward tumor-friendly directions [56]. Combining the above analyses of AQP1 and AQP9, we may guess that AQP1 is the tumor suppressor gene of ccRCC while AQP9 is the proto-oncogene of ccRCC. Although both genes belong to the AQPs family, they may play diametrically opposite roles, which are interesting and worthy of further exploration.

Changes in the text (the Conclusions):

In this study, the Oncomine database and the UALCAN database were used to analyze the mRNA of 14 members of AQPs, and the prognostic values of AQPs in ccRCC were analyzed using Kaplan-Meier plotter and GEPIA database. The results showed that in ccRCC patients, the mRNA expression levels of AQP0/8/9/10 were up-regulated in ccRCC, while those of AQP1/2/3/4/5/6/7/11 were down-regulated in ccRCC. The clinical database showed that the high mRNA expression levels of AQP0/8/9 were significantly associated with poor OS. On the contrary, the high levels of AQP1/2/3/4/5/6/7/10/11, especially the high levels of AQP1/4/7 were correlated with better OS in ccRCC patients. Among them, AQP1/7/11 have similarities in the tissue structure and positioning, and they are all highly expressed in the proximal tubules of the kidney. Perhaps AQP7/11 can become a potential diagnostic index and therapeutic target for ccRCC after AQP1. As the only two independent prognostic factors, AQP1 and AQP9 have shown differential expressions in ccRCC and normal kidney tissues as well as significant immune cell infiltration, which indicates that AQP1 and AQP9 may be used as a new prognostic and diagnostic marker in ccRCC. At the same time, each AQPs member may exert its function through different signaling pathways. To further explore the role of AQPs in ccRCC, more refined mechanism research and big data clinical trials are needed. Our study comprehensively analyzed the transcriptomics characteristics and the prognosis value of AQPs in ccRCC. Based on the verification of a large amount of data, the members of AQPs are expected to become new diagnostic, prognostic or therapeutic biomarkers markers for ccRCC targeted therapy.

Comments 11: Minor grammar errors throughout could use polishing by a language-fluent editor to showcase the work as effectively as possible, but it is understandable as written.

Answer 11: We sincerely appreciate your tolerance and understanding. We have used language editing software to check for grammatical errors that still exist in the manuscript and have made appropriate revisions.

References:

23. Beroukhim R, Brunet JP, Di Napoli A, Mertz KD, Seeley A, Pires MM, et al. Patterns of gene expression and copy-number alterations in von-hippel lindau disease-associated and sporadic clear cell carcinoma of the kidney. Cancer Res. 2009;69(11):4674-81.

24. Gumz ML, Zou H, Kreinest PA, Childs AC, Belmonte LS, LeGrand SN, et al. Secreted frizzled-related protein 1 loss contributes to tumor phenotype of clear cell renal cell carcinoma. Clin Cancer Res. 2007;13(16):4740-9.

25. Lenburg ME, Liou LS, Gerry NP, Frampton GM, Cohen HT, Christman MF. Previously unidentified changes in renal cell carcinoma gene expression identified by parametric analysis of microarray data. BMC Cancer. 2003;3:31.

26. Yusenko MV, Kuiper RP, Boethe T, Ljungberg B, van Kessel AG, Kovacs G. High-resolution DNA copy number and gene expression analyses distinguish chromophobe renal cell carcinomas and renal oncocytomas. BMC Cancer. 2009;9:152.

27. Jones J, Otu H, Spentzos D, Kolia S, Inan M, Beecken WD, et al. Gene signatures of progression and metastasis in renal cell cancer. Clin Cancer Res. 2005;11(16):5730-9.

28. Higgins JP, Shinghal R, Gill H, Reese JH, Terris M, Cohen RJ, et al. Gene expression patterns in renal cell carcinoma assessed by complementary DNA microarray. Am J Pathol. 2003;162(3):925-32.

33. Huang Y, Murakami T, Sano F, Kondo K, Nakaigawa N, Kishida T, et al. Expression of aquaporin 1 in primary renal tumors: a prognostic indicator for clear-cell renal cell carcinoma. Eur Urol. 2009;56(4):690-8.

56. Jing J, Sun J, Wu Y, Zhang N, Liu C, Chen S, et al. AQP9 Is a Prognostic Factor for Kidney Cancer and a Promising Indicator for M2 TAM Polarization and CD8+ T-Cell Recruitment. Front Oncol. 2021;11:770565.

76. Maynard MA, Ohh M. Von Hippel-Lindau tumor suppressor protein and hypoxia-inducible factor in kidney cancer. Am J Nephrol. 2004;24(1):1-13.

---

## [Decision Letter · Decision Letter 2]

14 Feb 2022

Abnormal expression and the significant prognostic value of aquaporins in clear cell renal cell carcinoma

PONE-D-21-25893R2

Dear Dr. Chong,

We’re pleased to inform you that your manuscript has been judged scientifically suitable for publication and will be formally accepted for publication once it meets all outstanding technical requirements.

Kind regards,

Graça Soveral, PhD

Academic Editor

PLOS ONE

Additional Editor Comments (optional):

Reviewers' comments:

Reviewer's Responses to Questions

**Comments to the Author**

1. If the authors have adequately addressed your comments raised in a previous round of review and you feel that this manuscript is now acceptable for publication, you may indicate that here to bypass the “Comments to the Author” section, enter your conflict of interest statement in the “Confidential to Editor” section, and submit your "Accept" recommendation.

Reviewer #2: All comments have been addressed

2. Is the manuscript technically sound, and do the data support the conclusions?

Reviewer #2: Yes

3. Has the statistical analysis been performed appropriately and rigorously? 

Reviewer #2: Yes

4. Have the authors made all data underlying the findings in their manuscript fully available?

Reviewer #2: Yes

5. Is the manuscript presented in an intelligible fashion and written in standard English?

Reviewer #2: Yes

6. Review Comments to the Author

Reviewer #2: The work provides a comprehensive analysis of patterns of AQP expression in RCC based on extensive database mining. The major concerns raised in my prior review have been addressed. The writing style could be improved but is acceptable. A minor detail is to remove the double listing of the antibody used for AQP12A & B in the Methods, lines 177-178.

7. PLOS authors have the option to publish the peer review history of their article (what does this mean?). If published, this will include your full peer review and any attached files.

Reviewer #2: **Yes: **Andrea Yool

---

## [Editor Report · Acceptance letter]

24 Feb 2022

PONE-D-21-25893R2 

Abnormal expression and the significant prognostic value of aquaporins in clear cell renal cell carcinoma 

Dear Dr. Chong:

I'm pleased to inform you that your manuscript has been deemed suitable for publication in PLOS ONE. Congratulations! Your manuscript is now with our production department. 

Kind regards, 

on behalf of

Dr. Graça Soveral 

Academic Editor

PLOS ONE